# Epigenomic and Metabolomic Integration Reveals Dynamic Metabolic Regulation in Bladder Cancer

**DOI:** 10.3390/cancers13112719

**Published:** 2021-05-31

**Authors:** Alba Loras, Cristina Segovia, José Luis Ruiz-Cerdá

**Affiliations:** 1Unidad Mixta de Investigación en TICs Aplicadas a la Reingeniería de Procesos Socio-Sanitarios (eRPSS), Universitat Politècnica de València-Instituto de Investigación Sanitaria La Fe, 46026 Valencia, Spain; 2Epithelial Carcinogenesis Group, Centro Nacional de Investigaciones Oncológicas (CNIO), 28029 Madrid, Spain; 3Unidad Mixta de Investigación en Nanomedicina y Sensores, Universitat Politècnica de València-Instituto de Investigación Sanitaria La Fe, 46026 Valencia, Spain; Jose.L.Ruiz@uv.es; 4Servicio de Urología, Hospital Universitario y Politécnico La Fe, 46026 Valencia, Spain; 5Departamento de Cirugía, Facultad de Medicina y Odontología, Universitat de València, 46010 Valencia, Spain

**Keywords:** bladder cancer, metabolic pathways, metabolism, metabolomics, epigenetics, biomarkers, miRNAs

## Abstract

**Simple Summary:**

Current diagnostic and follow-up methods for the clinical management of bladder cancer (BC) have limitations, and there is an urgent unmet need for non-invasive biomarkers for this highly prevalent disease. Furthermore, personalized treatments for patients could improve their quality of life and overall survival. The aim of this article is to review the literature in this area, with a primary focus on metabolic and epigenetic biomarkers of BC, as well as the targeted therapies discovered to date. We show the dynamic biological interplay established between epigenomics and metabolomics in the context of BC. These findings may be useful both for researchers and physicians in the field of BC, and could facilitate clinical decision-making regarding patients at diagnosis, prognosis, monitoring, or treatment.

**Abstract:**

Bladder cancer (BC) represents a clinical, social, and economic challenge due to tumor-intrinsic characteristics, limitations of diagnostic techniques and a lack of personalized treatments. In the last decade, the use of liquid biopsy has grown as a non-invasive approach to characterize tumors. Moreover, the emergence of omics has increased our knowledge of cancer biology and identified critical BC biomarkers. The rewiring between epigenetics and metabolism has been closely linked to tumor phenotype. Chromatin remodelers interact with each other to control gene silencing in BC, but also with stress-inducible factors or oncogenic signaling cascades to regulate metabolic reprogramming towards glycolysis, the pentose phosphate pathway, and lipogenesis. Concurrently, one-carbon metabolism supplies methyl groups to histone and DNA methyltransferases, leading to the hypermethylation and silencing of suppressor genes in BC. Conversely, α-KG and acetyl-CoA enhance the activity of histone demethylases and acetyl transferases, increasing gene expression, while succinate and fumarate have an inhibitory role. This review is the first to analyze the interplay between epigenome, metabolome and cell signaling pathways in BC, and shows how their regulation contributes to tumor development and progression. Moreover, it summarizes non-invasive biomarkers that could be applied in clinical practice to improve diagnosis, monitoring, prognosis and the therapeutic options in BC.

## 1. Introduction

BC is the second most common urological malignancy after prostate cancer, and its development has previously been shown to be strongly related to smoking, schistosomiasis infection, and occupational exposure to certain chemicals [1,2]. Worldwide, BC represents the sixth most frequent tumor with 424,082 new cases per year, and it is considered within the ten deadliest cancers [3].

According to histological criteria, 75% of newly diagnosed BCs are non-invasive (non-muscle-invasive BCs; NMIBCs) and have a 70% risk of recurrence and a 20% risk of progression, despite being treated with surgery (transurethral resection (TUR) of tumor), local chemotherapy, or non-specific immunotherapy (Bacillus Calmette–Guérin (BCG)) [1,4,5]. The remaining 25% of BCs are muscle-invasive BCs (MIBCs) and require radical cystectomy, usually followed by cisplatin-based chemotherapy. Patients with a poor performance status and/or metastatic disease have limited treatment options but may benefit from novel therapies such as immunotherapy, e.g., those that have recently been approved by the United States Food and Drug Administration (FDA) [6].

The clinical management of BC is complex. Macroscopic or microscopic urinary hematuria is one of the most prevalent symptoms in early stage BC, but alone it has low specificity (5%) since it can be present in other benign pathologies such as cystitis or urinary tract infections [7]. Therefore, urinary cytology and cystoscopy are routinely used for BC diagnosis and follow-up. Urinary cytology is a non-invasive procedure with a reasonable in-house cost, but it has poor sensitivity in low-grade BC detection. Consequently, white light cystoscopy is the gold standard for BC detection. However, this method also has drawbacks related to the omission of carcinomas in situ and preneoplasic lesions, user-dependent interpretation, invasiveness, and high costs [7,8,9,10]. In fact, the clinical management of NMIBC is one of the most expensive due to lifelong patient monitoring through cystoscopy and urinary cytology [11] to control the appearance of tumor recurrences and progression, and also due to the TUR which is carried out in recurrent BC. Due to the fact that urinary cytology and cystoscopy cannot provide prognostic information on BC disease development, the European Organization for the Research and Treatment of Cancer (EORTC) criteria are used to stratify NMIBC patients into low, intermediate, or high-risk groups, which are related to disease recurrence or progression [12]. However, although the EORTC scoring system is useful to guide the treatment of patients, it is obtained by the combination of static parameters (e.g., tumor grade and stage, number of tumors, size of tumors, presence of CIS) that do not reflect the dynamic behavior of tumors. 

Taking into account the shortcomings of cystoscopy and urine cytology, research is being conducted to find specific and non-invasive BC biomarkers that provide dynamic information of tumors and improve patient management by avoiding unnecessary cystoscopies in the surveillance of them. 

In the last decade, liquid biopsies have revolutionized oncology as a novel, non-invasive method to evaluate the treatment responses, assess therapy resistance, or characterize the tumor phenotype. Biomolecules such as circulating tumor cells (CTCs), circulating cell-free tumor deoxyribonucleic acid (ctDNA), messenger ribonucleic acids (mRNAs), micro-RNAs (miRNAs), long non-coding RNAs (lncRNAs), proteins and peptides, metabolites and vesicles (exosomes and endosomes) can be obtained from liquid biopsies and analyzed to provide information about the tumor [13]. Among samples used to find BC biomarkers, urine and blood have been the most frequent since the bladder releases cells and molecules into these biofluids [14]. To date, different assays based on CTCs, sediment cells, proteins, and mRNA detection have been carried out in urine or serum, and have received FDA approval for BC diagnosis and/or follow-up. Some examples are uCyt+, UroVysion, UroMark, CellSearch, CxBladder, CxBladder Monitor, Xpert BC Detection, NMP22, BTA TRAK and BTA stat [15]. However, given their low sensitivities and/or specificities, none of them have been shown to be superior to cystoscopy and have thus not yet been implemented into clinical practice. Currently, liquid biopsy is a promising non-invasive biomarker approach, and thus it may improve the management of BC.

Among molecular and analytic techniques used to identify biomarkers, metabolomics and epigenomics have developed rapidly in the past decade. Metabolic reprogramming and epigenetic modifications are two well-known hallmarks of cancer and their regulation is tightly linked to the tumor microenvironment and the eenvironment (e.g., microbiota), but is also influenced by other molecular processes (i.e., the genome, transcriptome, and proteome) [16,17]. Consequently, metabolomics and epigenomics have been found to be dynamic and closely reflect the phenotype of the tumor [18,19]. In addition, several studies have shown that the metabolome and epigenome establish bidirectional relationships in cancer cells. The metabolic reprogramming of cancer cells supports bioenergetic and biosynthetic demands of proliferation, but also alters the epigenetic landscape by modulating epigenetic metabolites. Furthermore, epigenetic mechanisms regulate metabolic gene expression to offer adaptive responses to rapidly changing environmental conditions and prolong tumor cell survival [20,21]. 

Understanding this dynamic relationship between metabolism and epigenetics and how they may be dysregulated in cancer is crucial to identify novel therapeutic targets and biomarkers. Additionally, it may provide a better understanding of the biological machinery underlying each tumor phenotype, thereby taking one step closer to precision medicine for the individualized treatment of patients in different types of cancer. 

The review presented here is, to our knowledge, the first that specifically analyzes the interplay among epigenomics, metabolomics, and cell signaling pathways in the context of BC. It provides an overview of the complex interplay between these biochemical and molecular processes and highlights the metabolites, metabolic enzymes, miRNAs, and lncRNAs postulated as emerging clinical biomarkers and therapeutic targets in the context of BC. This review integrates the current knowledge about the pathology of BC and may help to improve clinical decision-making regarding BC patients, whether at the level of diagnosis, prognosis, monitoring, or treatment. 

## 2. Metabolic Rewiring Controls the Epigenome in BC

It is widely known that cancer development and progression are due to genetic mutations in DNA. However, the role of metabolism and epigenetics has only been recognized in the last decade when the reprogramming of energy metabolism and epigenetic plasticity have been identified as two emerging hallmarks of cancer [22,23]. 

Cancer cells alter their metabolic and nutrient uptake pathways during tumor initiation, growth, and metastasis through a tightly regulated program of metabolic plasticity. This allows them to sustain the energetic and biosynthetic demands of cell proliferation and to adapt to hostile and ever-changing environments [24]. Epigenetic modifiers act on metabolic gene expression to induce changes in biochemical pathways, and many of the chemical modifications in DNA and histones derive from intermediates of cellular metabolic pathways. This indicates that fluctuations in metabolic concentrations affect the deposition and removal of chromatin modifications. Emphasizing on this last issue, several mechanisms have to be considered, such as: (i) the alteration of specific metabolites’ concentrations that act as epigenetic cofactors or substrates; (ii) the generation of oncometabolites which act as inhibiting or activating epigenetic enzymes; and (iii) the translocation of metabolic enzymes and metabolites into the nucleus [20]. Below, we address these regulation processes in the context of BC.

### 2.1. Metabolites and DNA/Histone Methylation Processes

DNA methylation is one of the most studied epigenetic mechanisms in cancer, including BC. Methylation can be produced directly in promoter regions of cancer-related genes (CpG islands) or in residues of histones, and this can control DNA accessibility and regulate gene expression (see Section 3.1.1). In DNA or histone methylation processes, the availability of methyl groups is essential for the action of histone methyltransferases (HMTs) and DNA methyltransferases (DNMT) [20]. In this context, the methionine and folate cycles, as well as the metabolites involved in these pathways (serine, methionine, and the cofactor S-adenosyl-methionine (SAM)), have an important role in supplying one-carbon groups [20] and they are closely related to DNA methylation processes (Figure 1). High levels of these metabolites have been found in BC samples and are postulated as candidate biomarkers of BC [25,26]. SAM provides methyl groups that release S-adenosyl-homocysteine (SAH), an inhibitor of DNMTs and HMTs. Therefore, the SAM/SAH ratio is a major determinant of chromatin methylation. It is known that an increased SAM/SAH ratio correlates with hypermethylation of tumor suppressor genes and inappropriate silencing, whereas a decreased SAM/SAH ratio contributes to reduced methylation at the promoters of oncogenes (Figure 1) [27].

Demethylation reactions are also susceptible to metabolic fluctuations of TCA cycle intermediates, such as α-KG, succinate, fumarate, and acetyl-CoA, which have been postulated as BC biomarkers [26,28]. They act on chromatin-modifying enzymes such as the 2-oxoglutarate-dependent dioxygenases (2-OGDO) family, which include ten eleven translocation (TET) enzymes, and the Jumonji (JHDMs) family of histone demethylases (Figure 1) [29,30,31]. These enzymes catalyze the hydroxylation and demethylation of proteins and nucleic acids and play an important role in epigenetic processes. α-KG acts as a positive cofactor of 2-OGDO, thus elevated levels of α-KG from glucose and glutamine catabolism would promote demethylation processes that would influence BC epigenetic landscapes by relaxing chromatin and activating oncogene expression. Additionally, α-KG is a substrate of prolyl hydroxylase (PHDs), a type of protein that regulates hypoxia-inducible factors (HIFs) [32,33]. HIF subunit alpha (HIF-1α) regulates various processes and, under hypoxic conditions, can promote cancer cell survival. In the presence of oxygen, PHD proteins hydroxylate proline residues on HIF-1α, which leads to HIF-1α ubiquitination by Von Hippel–Lindau tumor suppressor protein (pVHL) and its proteasomal degradation [34]. In the context of cancer, most tumors have hypoxic regions and, in this case, HIF-1α is stabilized and triggers changes in glycolysis, nutrient uptake, waste handling, angiogenesis, apoptosis, and cell migration, which promote tumor survival and metastasis [34].

In BC, HIF-1α has a predictive and prognostic role; its overexpression is known to stimulate angiogenesis and can lead to a poor prognosis for patients [35]. HIF-1α is also closely linked to metabolism and regulates glycolysis, fatty acid, and amino acid pathways. HIF-1α increases glucose uptake by upregulating glucose membrane transporters (GLUT1 and GLUT3), which has been correlated with BC progression and poor overall survival [36] (Figure 1). 

Furthermore, HIF-1α upregulates lactate dehydrogenase A (LDHA) to promote lactate production, regenerates nicotinamide adenine dinucleotide (NAD+), and increases the transcription of lactate transporters such as monocarboxylate transporter 1 (MCT1) [37,38]. This, in addition to high levels of hexokinase (HK), promotes a glucose flux towards pyruvate and lactate generation [39]. Previous studies have shown that lactate levels are related to BC progression and invasive or metastatic BC is known to have higher levels of this metabolite [38,39]. These metabolic processes would be in concordance with the Warburg effect [40]. Regarding lipid metabolism, several studies in cancer, including BC, have shown that HIF-1α increases the availability of fatty acids by regulating the action of fatty acid synthase (FAS), increasing fatty acid transport and reducing fatty acid oxidation [41]. In addition, HIF-1α plays an important role in amino acid metabolism, particularly in glutamine availability (Figure 1). 

Cancer cells use glutamine as an energy substrate, a precursor of fatty acids, a donor of carbon and nitrogen for generating nucleotides or other amino acids, and to maintain the poll of intermediate metabolites such as acetyl-CoA or α-KG. These last metabolites are important for the anaplerotic reactions of the TCA cycle, but also for epigenetic processes by the effect that they have on HAT and HDM [34]. Due to the role that glutamine has in BC cells, several studies have investigated the inhibitory action that lncRNAs (e.g., lncRNA-p21) or miRNAs (e.g., miR-1, miR-1-3p, miR-9, miR-129) exert on GLS regulation (more details in Section 3.1.2) [42,43].

On the other hand, fumarate hydratase (FH) and succinate dehydrogenase (SDH) genes are mutated in many human cancers, including BC, which leads to the accumulation of their substrates, fumarate and succinate, respectively [30]. This is consistent with metabolomic analyses, which have shown increased levels of these metabolites in BC [28,44], but also with transcriptomic studies that have shown a strong deregulation of TCA cycle genes [14]. Among others, succinate, fumarate and α-KG are considered oncometabolites. This term refers to metabolites that are significantly elevated in tumor cells compared with control cells [45]. Succinate and fumarate, together with 2-hydroxyglutarate (2-HG), can inhibit PHDs activity under normoxic conditions [31]. Hence, succinate and fumarate could act as competitors of α-KG, inhibiting JHDMs and TET activity, and acting on bladder tumor biology through a profound impact on epigenetic effector activity [46,47,48] (see Figure 1). In conclusion, tumor-gene expression is regulated by epigenetic enzymes, the activity of which is dependent on metabolite availability (substrates). 

### 2.2. Metabolites, Histone Acetylation Processes and Sirtuins

Another important metabolite, acetyl-CoA, is synthetized in several metabolic pathways (mitochondria, cytosol, and nucleus) from several sources, namely pyruvate, acetate, fatty acid β-oxidation, and amino acid catabolism. Metabolomic studies have reported elevated levels of acetyl-CoA in BC [28], and have particularly highlighted the role of glutamine as a substrate for acetyl-CoA synthesis [14]. Additionally, upregulated expression of acetyl-CoA synthase enzymes, such as ATP citrate-lyase (ACYL) or acetyl-CoA synthetase short chain family (ACSS), has been frequently found in BC cells, and some studies have reported the importance of ACSS3 for histone acetylation [49]. Acetyl-CoA acts as a cofactor which modulates kinetic and binding parameters of histone acetyltransferases (HATs). Nevertheless, CoA, the product of histone acetylation reaction, acts as an inhibitor. Therefore, the acetyl-CoA/CoA ratio has been postulated as the most important regulator of the enzymatic activity and specificity of HATs, rather than the absolute levels of acetyl-CoA [50]. In brief, high intracellular acetyl-CoA levels would trigger histone acetylation, an epigenetic marker associated with open chromatin, activating oncogenes linked with BC progression, proliferation and migration [20,30]. 

Another connection between metabolic processes and histone acetylation is provided by sirtuins (SIRTs), a type of NAD+-dependent histone deacetylases (HDACs) [51]. The activity of these enzymes is closely linked with the NAD+/NADH ratio, and consequently with the energy status in the cell. For example, when glycolytic activity is enhanced, the NAD+/NADH ratio decreases, thereby inhibiting SIRT catalysis [20,30]. The low NAD+/NADH ratio, together with an increase in HATs activity by elevated acetyl-CoA levels, could contribute to histone hyperacetylation and therefore an aberrant gene expression in BC [30]. 

### 2.3. Role of Metabolites in the Nucleus

Finally, the translocation or production of commonly cytosolic metabolic effectors in the nucleus can supply essential intermediates to epigenetic machinery in specific chromatin regions, which affects gene expression. Increased SAM levels in the nucleus support epigenetic methyltransferase activity at specific regions of chromatin [20]. This has been observed in cancer cells and is related to the translocation of splicing variants of MATs (S-adenosylmethionine synthetase, also known as methionine adenosyltransferase). Upregulated MAT1A levels have been reported in BC, specifically after treatment with chemotherapy, so MAT1A and possibly SAM could be related to the repression of tumor suppressor genes, (e.g., whose inhibition could confer tolerance or resistance to chemotherapy). Conversely, increased nuclear levels of acetyl-CoA can be produced by free diffusion of citrate or acetyl-CoA, but also by transient localization of the enzymes involved in its synthesis: ACSS2, ACLY, pyruvate dehydrogenase complex (PDC), and CAT (carnitine acetyltransferase). Post-translational modification of these enzymes within the nucleus or their association with lysine acetyltransferases (KATs) and transcription factors would explain their roles in chromatin regulation [48]. When there is DNA damage, ACYL is phosphorylated within the nucleus, which promotes histone H4 acetylation near sites of DNA double-strand breaks to repair them. Therefore, in response to DNA damage, ACLY phosphorylation would be enhanced, which would allow an increase in the capture of citrate or acetyl-CoA in the nucleus [48]. On the other hand, ACSS2 is recruited to specific genomic loci to supply acetyl-CoA for site-specific histone acetylation. Some studies have found that ACSS2 is translocated to the nucleus under low-glucose conditions upon phosphorylation by AMPK. Since cellular acetyl-CoA levels decrease when glucose is limited, a localized source of acetyl-CoA generated by ACSS2 could ensure the availability of this metabolite to KATs for histone acetylation [48]. PDC acts as a co-activator of signal transducers and activators of transcription 5 (STAT5) proteins. STAT5 proteins regulate specific nuclear genes in response to growth factors and cytokines which are linked to crucial cellular functions such as proliferation, differentiation, and survival. The role of STAT proteins is underscored in the field of cancer because tumors have an aberrant constitutive activation of them, which significantly contributes to tumor cell survival and malignant progression of disease [49]. Specifically in the context of BC, the findings obtained by Sun Y et al. suggested that the inhibition of STAT signaling by diindolylmethane (DIM) could decrease the invasiveness of BC, since DIM induced apoptosis in radioresistant cell lines. Therefore, DIM plus radiotherapy could be useful in overcoming such resistance [50]. Other studies performed in BC cell lines using inhibitors against STAT3/5 such as Stattic, Nifuroxazide and SH-4-54 also showed reduced survival and increased apoptosis. In a xenograft model, Static monotherapy had effects on tumors, but its combination with chemotherapy had additive effects. These findings highlight that inhibitors against STAT3/5 are promising as novel mono- and combination therapies in BC [51].

On the other hand, the regulation of NAD+/NADH levels in the nucleus is guaranteed by the activity of glycolytic enzymes (e.g., LDHA and glyceraldehyde 3-phosphate dehydrogenase (GAPDH)) since mitochondrial and nuclear membranes are impermeable to these cofactors [20]. Within the nucleus, NADH could be implicated in regulatory processes associated with histone acetylation, which in turn would influence transcriptional activity.

Last, the role of pyruvate kinase embryonic isozyme M2 (PKM2) in BC has previously been highlighted. Monomeric PKM2 translocates into the nucleus where it functions as a protein kinase that phosphorylates histones during gene transcription and chromatin remodeling [52]. Additionally, PKM2 upregulates the expression of c-Myc and cyclin D1, promoting the Warburg effect and cell cycle progression, respectively. Therefore, the role of nuclear PKM2 has been described as crucial for tumorigenesis, angiogenesis, and metastasis, and this protein has been postulated as a target for treating human cancers, including BC [53]. Numerous studies have correlated PKM2 overexpression with the development and metastasis of BC through promoting cell proliferation, migration and invasion via the mitogen-activated protein kinase (MAPK) signaling pathway [54], but also with advanced BC chemoresistance to cisplatin [55] or anticancer efficiency to pirarubicin [56]. Consequently, PKM2 could be a potential molecular prognostic marker of BC [57]. 

In brief, metabolic enzymes in the nucleus link metabolic flux to gene regulation, and allow nuclear membrane-impermeable metabolites to be used in epigenetic processes [58]. This metabolism–epigenetics axis would facilitate the adaptation to a changing environment around bladder tumors, providing a potential novel therapeutic target. The role that metabolites can play in modulating epigenetic enzyme action and gene expression is depicted in Figure 1.

## 3. Epigenetics Control Metabolic Reprogramming

Epigenetic regulation of gene expression is one of the most efficient stimulus response mechanisms. Extrinsic and intrinsic signals shape the plasticity of tumor cells to allow them to adapt to rapidly changing environmental conditions. These signals drive tumor metabolism [59], which can influence epigenetic mechanisms in several ways, such as the control of metabolite concentration necessary as cofactors or substrate for epigenetic enzymes, or the control of oncometabolites which regulate the expression of different epigenetic enzymes, among others, which have been explained above. These signals also drive metabolic reprogramming through changes in epigenetic modification patterns, achieving a rapid and coordinated response under unfavorable conditions for survival [27].

The functions of the epigenome are fundamental for the normal status of gene expression, and its alterations affect basic cellular processes such as proliferation, apoptosis or differentiation [60,61]. Epigenetics is defined as the heritable changes that occur in gene expression that do not involve alterations in the nucleotide sequence of DNA. These changes are basically divided into DNA methylation and modifications of the histone tails that allow the opening or closing of the chromatin. DNMT enzymes produce an irreversible silencing in the heterochromatin (closed chromatin state) and are divided in two groups: those involved in maintaining the methylation pattern in each cell replication (DNMT1) and those charged with *de novo* methylation (DNMT3a and DNMT3b). Chromatin regulatory elements are small molecules that regulate dynamic and reversible processes based on small post-translational modifications (PTMs) such as acetylation or methylation, among others, that make up the histone code. PTMs can be written by methyltransferases (HMTs) or acetylases (HATs), they can be deleted by demethylases (HDMs) and deacetylases (HDACs), and they can be read by different effector molecules to direct a particular transcriptional result. The main gene transcription marks are the acetylation of histones 3 and histone 4 (H3Kac, H4Kac) and methylation of histone 3 in lysine 4, 36 and 79 (H3K4me, H3K36me, H3K79me), while the best known gene repression marks are histone 3 methylation in lysines 9 and 27 and histone 4 in lysine 20 (H3K9me, H3K27me, H4K20me) [52]. The role of epigenetics is broad; it not only involves chromatin modifiers or changes in DNA methylation, but also includes non-coding RNA (ncRNA) expression (miRNA and lncRNA) that works in coordination with chromatin remodeling complexes and regulates the expression of multiple genes [62]. The metabolic reprogramming affects pivotal biological processes of the tumor cell such as survival, proliferation, or migration, among others. All these processes are the product of obtaining energy and its use in different chemical metabolic reactions. These reactions and the regulation of the use of energy to sustain demand underlie an aberrant epigenetic regulation of genes which are part of the main tumor metabolic routes and different oncogenic signaling pathways. Therefore, metabolism modulation can occur by epigenetic dysregulation of DNA methylation, histone modifications and ncRNAs.

The interplay of epigenetics and metabolic rewiring is complex. The epigenetic mechanisms, whether direct or indirect, that control metabolism are multiple and not all are well defined. However, there are two key connections between epigenetics and tumor metabolism: (i) epigenetic alterations which are directly related to the expression of metabolic enzymes; and (ii) epigenetic alterations that indirectly influence the signaling transduction cascades involved in the control of cellular metabolism.

Epigenetic alterations can also regulate major signaling transduction cascades. Many of them are well-known oncogenic pathways in cancer, especially in BC, which is considered an epigenetic disease. Chromatin remodeling gene mutations are more frequent in BC than in any other solid tumor. Importantly, they seem to be highly altered in the MIBC, where at least 89% of the alterations are in histone-modifying genes and 64% in genes associated with nucleosome positioning [63,64].

In recent studies, enhancer of zeste homolog 2 (EZH2), the main enzyme of the gene repressor complex Polycomb 2 (PRC2), has been shown to play an important role in tumor development and progression [65]. In NMIBC, EZH2 has been shown to predict recurrence and progression [66]. For example, EZH2 promotes changes in global gene expression, including aberrant expression of lncRNA HOTAIR, which acts in concert with EZH2 to mediate gene repression in high-risk tumors [67]. EZH2 also promotes the silencing of various miRNAs, such as the miR-200 family of miRNAs, which are involved in the repression of the epithelial–mesenchymal transition (EMT) and related to the increase in the probability of recurrence in the disease [68]. EZH2 is also known to cooperate with other modifiers such as DNMTs [69] and HDACs [70] to promote a permanent silencing of gene expression. A role of EZH2 in the activation of different oncogenic signaling pathways by turning off tumor suppressor genes has also been demonstrated. 

DNA methylation and epigenetic enzymes that are altered in BC define different tumor subtypes and can help in the diagnosis or prognosis of patients. Below, the main epigenetic alterations that are involved in metabolic rewiring, including those are known in BC, are shown.

### 3.1. Epigenetic Regulation of Metabolic Enzymes and Oncogenic Pathways in Metabolism 

There are different studies that show that various metabolic enzymes are altered by epigenetic events in tumor cells, as opposed to genetic mutations. These events can regulate tumor metabolic reprogramming directly or indirectly through DNA methylation, alterations in histone modification patterns, and by aberrant expression of non-coding RNAs (ncRNAs: miRNAs and lncRNAs; Figure 2), as will be shown throughout Section 3.1.1, Section 3.1.2, Section 3.1.3.

Additionally, metabolic reprogramming is activated by oncogenic signaling cascades, such as PI3K/Akt/mTOR signaling, transcriptional factors (TFs) such as HIF, MYC or p53, and the inactivation of tumor suppressor signaling, e.g., the LKB–AMPK pathway. Many of these are present in BC, such as PIK3 or genes which upregulate c-myc glycolysis genes [71]. The PI3K/Akt/HIF-1α axis mediates glycolysis and leads to autophagy through AMPK signaling in BC cells. Additionally, the lack of AMPK signaling increases mitochondrial ROS, which enhances HIF-1α signaling [71]. These cascades are in turn connected to many other factors that lead to a dynamic network and speak to the complex behavior and biology of the tumor. These signaling pathways and activation of TFs are closely related to tumor metabolism, but in turn can be regulated by epigenetic mechanisms and as it will highlight in relation to ncRNAs (Figure 2). The implications of these signaling pathways in metabolic rewiring will help to understand their epigenetic regulation. This is briefly described below within each section. 

AKT is known as the major regulator of glucose uptake and improves glucose metabolism via glycolysis and the pentose phosphate pathway [72]. MYC is usually induced by the Wnt/β-catenin, MAPK/ERK, and PI3K signaling pathways. In general, MYC induces the transcription of genes involved in glycolysis and glutaminolysis and contributes to control over redox balance [73]. MYC can also contribute to the supply of glycolysis intermediaries to the PPP for the biosynthesis of other molecules [73]. MYC can also regulate the use of glutamine by facilitating the activation of the expression of its transporters [74] or even controlling the repression of ncRNA targeting the enzyme GLS [75]. Thus, tumors that present MYC as a driver have a strong dependence on glucose or glutamine, which may indicate them as promising targets for the study of metabolic inhibitors. The classic tumor suppressor TP53 also has an emerging role in metabolic reprogramming. The activation of p53 represses the transcription of GLUT1, GLUT3 and GLUT4 transporters and activates the transcription of proteins involved in the electron transport chain for their replacement [76], and can even bypass glucose through the PPP [77]. In addition, those tumors that show a loss of p53 functionality will show a glycolytic metabolic phenotype. In hypoxic conditions, higher levels of HIF are detected. The HIFα subunits stabilize and activate the transcription of numerous glycolytic enzyme genes [78], or lactate synthesis enzyme genes, and this has been demonstrated in BC [79]. HIF can also be constitutively activated under normoxic conditions by oncogenic pathways such as PI3K/Akt/mTOR [80,81] or through the inactivation of LKB1–AMPK signaling [78]. 

#### 3.1.1. DNA Methylation

DNA methylation is one of the most studied epigenetic mechanisms in cancer. Promoter hypermethylation usually occurs in tumor suppressor genes, DNA repair genes, cell cycle control, and invasiveness genes, and the silencing of their transcription causes cancer development and progression [82]. Thus, hypermethylation is correlated with the grade and stage of the tumor, with low-grade tumors being the least altered (10%) compared to high-grade (20%) and invasive tumors (30%) [83,84]. For example, the hypermethylation of GATA2, TBX2, TBX3 and ZIC4 in NIMBC is associated with progression to MIBC [84,85], but their connection with metabolic pathways is still unknown. Additionally, DNA hypomethylation in tumors leads to genomic instability and the activation of proto-oncogenes [86] and increases the risk of BC [87,88]. 

Numerous studies have shown that DNA methylation is related to glycolysis and glucose consumption. Indirectly, DNA methylation contributes to increased glucose uptake by silencing genes associated with glucose transporter degradation pathways such as GLUT1, allowing their overexpression [89]. Tumor suppressors, such as PTEN or LKB1, inhibit oncogenic signaling, which are central activators of glycolysis (Akt, AMPK, HIF and p53), and undergo hypermethylation of their promoters, facilitating the activation of glycolysis and the synthesis of macromolecules, thus helping to maintain the glycolytic phenotype [21,90]. Furthermore, the hypermethylation of certain metabolic enzyme genes has a direct impact on glycolysis. For example, the FBP1 gene (fructose 1,6-biphosphatase), which codes for one of the main enzymes of gluconeogenesis, shows hypermethylation of its promoter in different tumors, facilitating glycolysis [21,73,90]. In addition, the hypomethylation of promoters can activate the transcription of genes that code for glycolytic enzymes. Enzymes such as PKM2 or HK2 undergo promoter hypomethylation which allows them to be expressed and increases their availability, which leads to an accelerated glycolytic flow [91,92].

Moreover, recent studies link DNA methylation sites to fatty acid metabolism in BC [93,94,95,96,97]. TIMP3 promotor hypermethylation has been described in the regulation of lipid metabolism, fatty acid oxidation and cholesterol homeostasis in response to oxidative stress [96,97]. GSTP1 (glutathione S-transferase 1) is known to be hypermethylated in BC and this enzyme is involved in xenobiotic metabolism [98,99] and can regulate glycolytic and lipidic metabolism energetics, as well as oncogenic signaling pathways in other tumors [100]. Further, human GSTM1, part of the GST superfamily, has been reported to be transcriptionally downregulated by DNA methylation in BC [101]. Some cancer cells, including BC, overexpress DNMT1, DNMT3A, and DNMT3B, which in turn leads to DNA hypermethylation of promoter regions and the silencing of tumor suppressor genes [102,103]. This DNA methylation pattern commonly negatively affects gene expression, promoting tumor growth and progression and predicting therapy outcomes [104]. In fact, the main difference between low- (LG) and high-grade (HG) BC is related to the amount of aberrant hypermethylation in specific loci, with HG BC having a greater percentage of aberrant DNA methylation patterns (over 30%) when compared to LG BC [105,106]. Elevated FHIT, CDH1, CDH13, RASSF1A and APC promoter methylation levels correlate with poor prognosis, adverse clinicopathological features, BC progression, and reduced overall survival [107].

ASS1 and SAT1 genes, which are related to amino acid metabolism, have been shown to be hypermethylated in cancer, and their association with the rewiring of cisplatin-resistant BC has also been demonstrated [108]. In patients diagnosed with BC, the ABCB1/MDR1 drug transporter exhibits a dynamic degree of methylation, changing from a hypermethylation state during carcinogenesis to a hypomethylated state during chemotherapeutic treatments. Thus, it is a possible prognostic factor for disease recurrence and treatment response in BC [109].

Finally, it should be noted that, to date, approximately 90% of identified genes involved in drug metabolism and transport that are epigenetically regulated implicate DNA methylation. Therefore, further research in this area may facilitate our understanding of the multidrug resistant response in different patients with different types of cancers [110]. Cytochrome P450 enzymes can be hypomethylated and transcriptionally activated and the metabolic response to drugs can be improved, thereby promoting resistance to treatments [110,111]. Modifications such as histone methylation or acetylation frequently work in combination to mediate the DNA methylation status of genes related to drug metabolism [112,113]. Therefore, both DNA hypomethylation and hyper methylation can contribute to the glycolytic phenotype in tumor cells and to the activation of xenobiotic metabolism.

#### 3.1.2. Histone Modifications

Post-transcriptional modifications of histone tails regulate chromatin structure for gene activation or repression, replication, and DNA damage repair. The best studied and central modifications of epigenetic regulation include tailing amino acid methylation and acetylation marks, which are controlled by enzymes that can write them (HMTs or HATs) or can erase them (KDMs or HDACs).


**Histone Methyltransferases (HMTs)**


Currently, further research is still required to understand how histone modifications can control metabolic reprogramming. However, taking into account that histone methylation markers are the main regulators of gene transcription, it is highly likely that they modulate metabolism. HMTs control the methylation status of histones through repressor labels such as H3K9me, H3K27me3 and H4K20me3 [114,115], and there is evidence that some of these enzymes interfere in some way with cellular metabolic activities.

As mentioned above, EZH2 promotes global changes in gene and ncRNA expression, can cooperate with other chromatin remodelers or may co-occupy several gene loci with G9A in order to maintain gene silencing in a cooperative and coordinated manner, which has been shown in BC [116] and could explain some oncological properties of EZH2 that could favor heterogeneity in tumors [117]. As the master epigenetic regulator, EZH2 controls gene silencing through its mark H3K27me3 and it can alter the metabolic profile of tumor cells through the metabolism of glucose, lipids and amino acids, as demonstrated in recent studies [118]. EZH2 activity is affected by the availability of SAM from altered metabolic profiles, and, through its involvement in multiple metabolic pathways, EZH2 can also increase SAM gene and protein expression. Those events establish a positive feedback loop which improves the activity of EZH2 and favors tumor progression [118,119]. Furthermore, EZH2 is also influenced by the production of other metabolites involved in post-transcriptional modifications such as phosphorylation, acetylation or O-GlcNAcylation (O-linked N-acetylglucosamine modification) that can regulate the activity and stability of this enzyme [120,121].

EZH2 facilitates glucose metabolism in tumor cells in several ways. It can silence the transcription of HIF1α-directed hypoxia signaling repressors, which induces the transcription of metabolic genes such as *GLUT1, PDK*, or *HK2* and contributes to maintaining the Warburg effect [122]. EZH2 can also promote and regulate lipid metabolism by silencing WNT signaling pathways and overexpressing lipogenic genes such as *PPAR-γ* [123,124]. Additionally, TERT and EZH2 cooperate in the activation of PGC-1α, which is involved in the expression of FAS [125]. It has been shown that FAS is involved in the synthesis of triglycerides, and its expression is upregulated by SIRT6, a protein deacetylase known to promote adipogenesis and be repressed by EZH2 [126]. These studies demonstrate the participation of EZH2 in lipid synthesis; however, it is a field that still requires further research, especially due to the important role of EZH2 in BC. EZH2 can also regulate amino acid metabolism through multiple pathways, mainly contributing to the production of methionine for SAM which, in addition to enhancing the expression of EZH2, can affect amino acid transporters [127]. EZH2 also regulates the expression of transamination enzymes that participate in the production of α-KG to obtain glutamate, and EZH2 inactivation can upregulate glutamine metabolism [128]. New findings demonstrate that aldehyde oxidase (AOX1) is epigenetically silenced through EZH2 during the progression of advanced BC. *AOX1* silencing reconnects the tryptophan–kynurenine pathway, raising NADP levels that can increase metabolic flux through the PPP, allowing greater nucleotide synthesis [128,129]. Thus, AOX1-associated metabolites have a high predictive value for these tumors that do not have effective therapeutic opportunities.

EZH2 expression can be regulated at multiple levels. It can be transcriptionally induced by the activation of c-MYC or loss of p53 [130]. MYC regulates EZH2 directly by interacting with its transcriptional promoter or indirectly by controlling the repression of some miRNAs that silence EZH2 [131]. However, it can also be post-transcriptionally regulated by interaction with ncRNAs [132] or by the activation of signaling cascades such as PI3K–Akt [120]. 

Another important methyl transferase that is involved in amino acid metabolism is G9A. This enzyme can write the repression marks H3K9me1 and H3K9me2. Ribosome biogenesis and cell proliferation depend on the availability of serine, and G9A is known to increase glycolytic flux towards serine–glycine synthesis which is observed in BC [133]. G9A is overexpressed in many tumors, including BC [116], and can cooperate with TFs or with demethylases, such as KDM4C/JMJD2C, to maintain the H3K9me1 mark in the promoters of serine pathway-related genes to promote their transcriptional activation, including those for amino acid synthesis and transport [134,135]. Furthermore, like EZH2, G9A regulates transamination enzymes whose expression is activated due to H3K9 demethylation or G9A repression, and helps the production of other precursor metabolites such as NADH or α-KG, which have critical roles in the control of cellular metabolism related to cell proliferation and survival [135]. Patients with this type of epigenetic profile, as in the case of EZH2, are not suitable candidates for epigenetic inhibitor therapies as they could contribute to the metabolic reprogramming of tumors.


**Histone Acetyltransferases (HATs)**


The acetylation of histone lysine residues is established by HAT activity, using acetyl-CoA as an acetyl donor [136]. Therefore, tumors with high production of acetyl-CoA can destabilize acetylation levels. There are not many reports on the involvement of these enzymes in metabolic reprogramming, but it is known that the acetylation of PKM2 at the end of glycolysis decreases its activity, thus favoring the trafficking of intermediates for the biosynthesis of nucleic acids, amino acids and lipids [137], and that low levels of H4K16ac/H3K9ac are associated with a highly proliferative tumor profile, thus it is likely that these marks are related to metabolic rewiring [115,138], and would be very interesting to research.

Histone acetylation is a highly dynamic process being regulated by two enzyme family members, operating in an opposite fashion: the HATs and the HDACs. HDACs are overexpressed in various tumors, including BC, and histone acetylation levels decrease during progression to MIBC. Several studies indicate that global levels of histone acetylation are suitable biomarkers for patients with urological malignancies. For example, histone acetylation levels could be helpful to identify patients with understated pT1 tumors after TUR, identifying those who need cystectomy [139], or could help to identify how patients will respond to treatment with HDAC inhibitors because the use of these therapies results in decreased global histone acetylation levels and a poor prognosis outcome for patients [139,140].


**Histone Lysine Demethylases (KDMs)**


Lysine demethylases (KDMs) are often overexpressed and activated in solid tumors. These enzymes remove a methyl group from the histone tails by oxidation in a flavin adenine dinucleotide (FAD) or α-KG-dependent manner [141].

Several KDMs play an active role in metabolic rewiring in cancer, for example, KDM3A/JMJD1A, which promotes BC progression by enhancing glycolysis through the coactivation of HIF-1α [142]. KDM3A demethylates glycolytic gene promoters including GLUT1, HK2, phosphoglycerate kinase 1 (PGK1), and LDHA, among others, through H3K9me2 mark, leading to their transcriptional activation [142]. Additionally, increased H3K27ac binding on HIF-1α induces GLUT3 overexpression through KDM3A binding, further contributing to BC’s glycolytic phenotype [143]. Finally, it is notable to mention that the role of KDM6A/UTX, an enzyme that catalyzes the demethylation of H3K27me2 and H3K27me3, acts as a tumor suppressor and can interact with other epigenetic elements.

Regarding glutaminolysis, the production of α-KG is reduced under hypoxic conditions because it depends on the available local levels of glutamine. Likewise, α-KG is a target for various histone demethylases [144]. The loss of KDM6A reproduces the effects of low glutamine levels, suggesting that histone demethylases may be dependent on α-KG, accentuating metabolic reprogramming [145]. It also causes important changes in the levels of H3K4me1/H3K27ac of enhancer TFs and allows methyl transferases, such as EZH2, to rewire H3K27me3 levels on the UTX-EZH2 target genes, e.g., the repressor genes of c-MYC [144] or IGFBP3, whose decreased levels are involved in glucose metabolism [146]. KDM6A is one of the most frequently mutated enzymes in BC [147]. The loss of KDM6A promotes enrichment in PRC2-regulated signaling and confers specific vulnerability to EZH2 inhibition by converting tumoral cells into inducible synthetic lethality therapeutic targets and provides a new possibility of personalized treatments in urothelial tumors [148]. Although the specific role of KDM6A in metabolic rewiring in BC is unknown, a correlation has been observed between urothelial tumors with altered KDM6A and the upregulation of DNA repair genes and mTORC1 signaling, which stimulates aerobic glycolysis and lipid and nucleotide synthesis [147,149]. Therefore, KDM6A is a fundamental part of epigenetic regulation, making it a potential candidate involved in metabolic processes.


**Histone Deacetylases (HDACs)**


HDACs are responsible for removing acetyl groups and are categorized into four classes. HDAC Class III, or SIRTs, has been the most extensively studied concerning roles in cell metabolism regulation [21,27,90]. SIRTs act on the activity of TFs implicated in the transcription of genes involved in glycolysis, gluconeogenesis and lipid metabolism [150]. Dysregulated expression of various HDACs has been described in urothelial tumors [151]. These enzymes could deacetylate lysines such as H3K27 to be subsequently methylated by EZH2, and even PRC2 could recruit HDACs through EED to direct cooperative gene repression [151].

SIRT1 is mainly involved in tumor suppressor in cancer, including BC [140,152,153]; it can suppress glycolysis indirectly through the deacetylation of HIF, which in turn can regulate the transcription of various glycolytic enzymes such as LDH, G6P, PFK-1, PGK-1, PGAM-1 or transporters such as GLUT1 or GLUT3 [154,155]. Moreover, SIRT1 can regulate gluconeogenesis and lipid metabolism [150].

SIRT2 is also considered a BC tumor suppressor, and can participate in metabolic dysregulation indirectly by stabilizing MYC via deacetylation of a repressor, which functions as a positive feedback loop [156]. It also contributes to gluconeogenesis by deacetylation and activation of phosphoenolpyruvate carboxykinase (PEPCK) in the absence of glucose [150]. SIRT2 silencing induces the inhibition of the HDAC6 family member, causing a significant suppression of BC cancer cell migration and invasion. This strongly supports the cooperative actions between SIRT2 and HDAC6 in urothelial malignancies [140].

SIRT3 is associated with glycolytic metabolism [157]. It can regulate glucose balance in a HIF1α-dependent manner at the mitochondrial level [158]. In contrast to SIRT3, SIRT4 is involved in the inhibition of GDH, repressing glutamine metabolism [159]. In addition, it is also able to act on the activity of pyruvate dehydrogenase that catalyzes the conversion of pyruvate to acetyl-CoA [150]. Of note, SIRT 3 and SIRT4 downregulation has been reported in BC [140,153].

SIRT6 is the most reported SIRT involved in tumor metabolic reprogramming. Its primary role is in the regulation of glucose homeostasis and lipid metabolism [150]. SIRT6 works as a suppressor by blocking the HIF-dependent glycolytic switch and MYC-dependent ribosomal biosynthesis and glutaminolysis [21]. For instance, it can directly repress the expression of glycolysis genes by the deacetylation of H3K9, such as GLUT1 [154]. Functional studies of SIRT6 in BC cell lines confirmed its role in inhibiting glycolysis [160]. It should be noted that, in addition to regulating glycolysis, SIRT6 also participates in gluconeogenesis and lipid metabolism [150]. Its over-expression can inhibit the proliferation of BC cells, while its expression decreases with the progression of BC from T2 to T4 stage [140]. Therefore, SIRT6 could be a promising druggable biomarker for BC, as would be the case of SIRT4. SIRT7 can interact directly with the MYC factor, repressing its function [161,162]. Like SIRT6 (H3K9), the repressive mark of SIRT7 (H3K18) opposes the transcription of MYC-dependent genes and can therefore regulate the metabolic alterations mediated by this factor [150,162]. SIRT7 is upregulated in many cancers, including BC [140]; however, it has been reported that in BC it may play a dual role depending on the context, suggesting that the functional importance of SIRTs may change throughout cancer progression [153]. It has been demonstrated that SIRT7 levels decrease significantly in MIBC, suggesting that SIRT7 may promote a more aggressive phenotype [152]. Although the mechanism of dysregulation of SIRT7 in BC and its putative implication in metabolic rewiring have not yet been adequately addressed, it is speculated that this could be due to epigenetic mechanisms that allow a plastic expression of SIRT7 in both carcinogenesis and tumor progression [152,153]. A lower expression of SIRT7 could be related to a positive regulation of TFs of EMT processes, such as Snail or HOTAIR, which would participate in the recruitment of EZH2 to specific genes [162]. It has been demonstrated that the expression of SIRT7 is regulated by miRNAs such as miR-125b, which in turn interacts antagonistically with the lncRNA MALAT1 in BC [163]. However, the findings on the miR-125b–MALAT1 interaction and its possible participation in metabolic regulation should be further studied and confirmed [164]. 

In summary of this section, this is a possible scenario of epigenetic–metabolic reprogramming in BC. On the one hand, EZH2 can interact with other chromatin remodelers such as G9a and DNMTs regulating the expression of tissue-specific gene sets in BC. EZH2 and G9a work in obtaining α-KG, in transamination reactions involved in amino acid and lipid synthesis, and promote glycolysis and the PPP. Moreover, DNMT can hypermethylate genes that enhance glycolysis, but also it plays an important role in xenobiotic metabolism. Therefore, a relationship with EZH2, and consequently with G9a, could open an exploration path in resistance to therapy and the use of combined therapies in BC. Interestingly, KDMs such as UTX or KDM3A cooperate with HIF-α and are associated with high levels of α-KG and glutamine availability, enhancing amino acid synthesis, PPP activation, glycolysis as well as lipid synthesis.

On the other hand, it has been proven that EZH2 can interact with PI3K signaling, one of the major regulators of glucose uptake that enhances glycolysis and PPP, but also with HIF, MYC and p53, either directly or indirectly. In turn, there appears to be evidence of EZH2 regulation by these potent metabolic rewires. It should be noted that EZH2 regulates the expression of HIF suppressors and that the role of sirtuins is basically to keep HIF or MYC repressed, as is the case with SIRT6 and SIRT7 in BC. There is evidence that SIRT6 may be silenced by EZH2 and that SIRT7 decreases as the disease progresses to MIBC. It can be assumed, then, that EZH2 could regulate sirtuin expression, together with ncRNAs, since EZH2 is involved in the recurrence and progression in BC. Thus, EZH2 not only controls gene silencing programs together with other chromatin remodelers, but can also regulate demethylation and deacetylation programs to orchestrate which metabolic pathways are enhanced for obtaining energy and precursors and which are maintained by the tumor cell. All this occurs through epigenetic regulation, one of the most fluid and dynamic machineries to obtain rapid responses to progress in the disease.

#### 3.1.3. ncRNAs

Recent studies have shown that ncRNAs regulate enzymes involved in metabolic pathways such as glycolysis and the mitochondrial TCA cycle, contributing to oncogenic metabolic programming. Their aberrant gene expression contributes to the establishment of diverse mechanisms that govern the plasticity of tumor cell metabolism [21]. These small RNA molecules are non-coding, but nevertheless have a regulatory role for gene expression at the post-transcriptional level [90]. The regulatory role that miRNAs and lncRNAs can exert on metabolic enzymes and glucose, lipid and amino acid metabolic pathways in BC is discussed below [165,166].

ncRNAs can actively regulate energetic signaling by targeting key metabolic transporters and enzymes, but also by directly or indirectly controlling the expression of tumor suppressors or oncogenes in different signaling pathways. 


**miRNAs**


There are numerous studies that link miRNAs with tumor metabolism, and they have functions in various types of cancer. miRNAs control crucial metabolic processes including glucose transport, glycolysis, the TCA cycle, glutaminolysis, altered lipid metabolism as well as amino acid biosynthesis [167], but they are also related to drug-metabolizing gene expression [106].

miRNAs can regulate the expression of numerous enzymes that participate in glucose uptake, including glycolytic enzymes such as HK2, controlled by miR-143, miR-145 and miR-155 in BC [168,169,170]; PKM2 regulated by miR-326 and miR133a/b [21]; or the expression of LDHA, which is controlled by miRNAs such as miR-34a or miR-200c in urothelial tumors, favoring the production of lactate [167,171], among others. However, they can also regulate the expression of glucose transporters. For instance, many miRNAs such as miR-199a, miR-138 or miR-150 can control the expression of GLUT1 [164]. Interestingly, miR-218 has been shown to regulate the expression of GLUT1, which leads to an enhancement of chemosensitivity to cisplatin in BC [172]. In addition, miR-93 and miR-133 regulate GLUT4 [173], and miR-195-5p or miR-106a improve the expression of GLUT3 in BC [174].

However, miRNAs arguably have the greatest impact on essential metabolic signaling pathways such as PI3K/Akt/mTOR and LKB1–AMPK [121,164], as well as the expression of TFs such as HIF, MYC and p53 that contribute to the metabolic phenotype of the tumor [21]. 

The expression of some miRNAs controls the activation of the PI3K/Akt/mTOR pathway [165]. miR-143-145 cluster and miR-133a regulate AKT expression in BC. Others regulate mTOR, such as miR-100 [175], or contribute to the inactivation of inhibitor phosphatases (PTEN), as in the case of miR-19a [176]. 

Regarding metabolism, miR-21 is a molecular switch of several aerobic glycolytic genes, such as GLUT1, GLUT3, LDHA, LDHB, PKM2, HK1 and HK2 in bladder tumors, and regulates glycolysis through the PI3K/Akt/mTOR pathway [177]. The AMPK pathway is one of the major sensors of cellular energy status that suppresses tumor growth during metabolic stress and is regulated by oncogenic miRNAs such as miR-451 or miR-33a/b [165]. The reactivation of LKB1–AMPK in BC cells improves apoptosis and autophagy [178,179]. However, the role of this pathway in metabolic reprogramming in BC requires further research.

These miRNAs, in turn, interact with other factors, e.g., HIF, to propagate hypoxia-induced signaling and its stabilization [180], and some miRNAs are induced by HIF1α in BC, such as miR-210, -193b, -125, or miR-145 [181]. p53 can also induce the expression of miRNAs with a partially suppressive role by inhibiting glycolytic enzymes like HK1 and HK2, such as miR-34a [165]. On the other hand, p53 is under the regulation of other oncogenic miRNAs such as miR-25, -30d, -33, -125b that can contribute to its stabilization [182]. 

Regarding the interaction of miRNAs with MYC, there is evidence that this factor can repress miRNAs with a suppressing role, such as miR-23a or miR-24b, or it can bind to the promoter of oncogenes of other miRNAs to promote the metabolism of glutamine [72,75].

miRNAs such as the miR-210 or miR-200 family of miRNAs can inhibit mitochondrial function by promoting glycolysis, glutaminolysis, and lactate production, which is crucial for adaptation to hypoxic microenvironments [92]. Altered expression of the miR-200 family has been reported in various cancers and is known to play an important role in BC [68]. This family includes five members located in two different clusters: miR-200a, miR-200b and miR-429 (cluster I) and miR-200c and miR-141 (cluster II). It has been shown that there is clear upregulation of cluster II in urothelial tumor tissue compared to normal tissue, and this overexpression may be mediated by the activation of specific oncogenic pathways such as MYC overexpression or p53 alterations [68]. Gene expression patterns show that miR-200 expression is involved in ncRNA metabolism and RNA splicing, and chromatin remodeling and histone modification processes are inversely related to miR-200 expression [68]. However, the expression levels of this family decrease as the disease progresses [183], suggesting a dual role of miR-200 in BC at different stages of the disease. The hypermethylation of CpG islands is known to induce high-stage miR-200 silencing in aggressive and infiltrating BC [184]. In addition, genes related to the expression of miR-200 significantly present the repressive mark H3K27me3, which indicates that the activity of EZH2 is inverse to the expression of these miRNAs and that EZH2 possibly participates in the repression of miR-200, which contributes to early recurrence, as we initially described before Section 3.1. It should be noted that studies relating to the participation of the members of this miRNA family in metabolic processes are scarce, even more so in BC, and that we have only found evidence regarding HIF-miR-200c and LDHA mechanisms [181].

miRNAs have been shown to target key enzymes involved in aerobic glycolysis, the TCA cycle and lipid metabolism [185], but also control the aberrant expression of central epigenetic enzymes, such as EZH2, and their expression can even be affected as a consequence of the hypermethylation of the gene that encodes them by both DNMT and HMT [186]. This means that in addition to metabolic dysregulation, ncRNAs further expand the intricate regulatory network in the mechanisms underlying tumorigenesis [90].

EZH2 is a direct target of miR-101 or miR-138 in BC [187]. However, these miRNAs also perform other functions. For example, in BC it has been shown that miR-101 can regulate the expression of cyclooxygenase-2, which is related to xenobiotic metabolism [188], and miR-138 in turn can control the expression of EMT factors such as ZEB2 [42].

Other miRNAs related to metabolic rewiring in BC are miR-1, miR-9 or miR-129. miR-1 acts as a tumor suppressor regulating GLS expression, which is crucial in glutamine metabolism [189]. miR-9 modulates the expression of the LASS2/CERS2 gene, which codes for a ceramide synthase, an enzyme involved in the metabolism of sphingolipids, which are part of cell membranes [190]. On the other hand, miR-129 may mediate the expression of GALNT1, an enzyme involved in protein metabolism [191].


**LncRNAs**


LncRNAs have been identified as important regulators of cellular metabolism. However, despite cumulative studies investigating altered expression profiles of lncRNAs during metabolic rewiring in cancer, the functional roles of these lncRNAs remain largely unexplored [192,193]. The expression pattern of an lncRNA can differ significantly depending on the metabolic process, making them crucial drivers of highly tissue-specific cancer phenotypes [194]. LncRNAs are involved in the regulation of oncogene expression, which induces metabolic reprogramming of HIF 1 α, c-MYC or p53 [195,196], but they can also interact with tumor suppressors, such as AMPK [197], oncogenic signaling pathways, metabolic enzymes or with other ncRNAs [193,194]. This network allows the maintenance of metabolic rearrangement during tumor response surrounding the microenvironment.

LncRNA-mediated glucose metabolism can occur through three different mechanisms: the alteration of the expression levels or distribution of GLUTs, the alteration of expression levels of glycolytic enzymes, or interactions with glycolytic genes and the modulation of their activity [194].

LncRNAs modulate glucose metabolism through glucose transporter regulation and glycolytic genes, and are implicated in many malignancies, including BC, such as ANRIL [198]. High tissue abundance of ANRIL in cancer is associated with aggressive clinicopathologic features, poor overall survival [199], and resistance to chemotherapy [200,201]. ANRIL interacts with signal transduction pathways in cancer such as PI3K/Akt/mTOR [202]. Furthermore, ANRIL promotes GLUT1 and LDHA expression, resulting in the upregulation of glucose uptake and the promotion of cancer progression via the Akt/mTOR pathway [194]. However, it can also interact with TFs such as c-MYC, which can transactivate ANRIL and promote tumor progression [202]. ANRIL also participates in the regulation of gene expression via mechanisms including chromatin modulation, TF binding, and miRNA regulation [203,204,205], and can be considered as a driver in cancer progression by increasing glucose uptake for glycolysis, and additionally, ANRIL has also been linked to fatty acid metabolism [206,207]. 

Other lncRNAs, such as HOTAIR and urothelial cancer-associated 1 (UCA-1), play an important role in BC. Previous data have shown that the expression of HOTAIR may be upregulated by EZH2 and that it is a predictor of disease recurrence and progression, and overall survival in MIBC [68]. Recently, its role in tumor glucose metabolism through the induction of GLUT1 gene expression has been discovered [208]. This lncRNA is also involved in OXPHO mitochondrial activity [209]. Although these metabolic roles have not been elucidated in BC, they could be part of the epigenetic regulation directed by an EZH2–lncRNAs axis that reinforces the reformulation of metabolism.

UCA-1 is the most studied lncRNA in BC. It promotes glycolysis by modulating HK2 via the activation of mTOR/STAT3 and miR-143 repression [129] and induces glutamine metabolism and redox regulation by targeting miR-16 in human BC [210]. Similarly to HOTAIR, UCA-1 has been related to mitochondrial activity [194], contributing to ARL2 induction through miR-195 inhibition in BC [192,211,212]. Therefore, UCA-1 could be related to metabolic readjustment in BC cells, although more in-depth investigations are necessary in order to prove this hypothesis. To date, it is known that UCA-1 shows association with HIF-1α in hypoxic environments [213], participates in the regulation of EMT processes though targeting miR-145–ZEB1/2 or miR-143/HMGB1 pathways [15,214,215], induces cisplatin/gemcitabine xenobiotic metabolism modulating miR-196a [216], and, curiously, it is under the regulation of miR-1 [217], one of the miRNAs involved in metabolic reprogramming in BC.

Lastly, although studies on lncRNAs and tumor metabolism in BC are scarce, it has been reported that lncRNA SLC16A1-AS1 promotes metabolic rewiring in BC disease progression [218]. This lncRNA creates an lncRNA–protein complex with E2F1, which facilitates its binding to two gene promoters: SLC16A1/MCT1, a monocarboxylate transporter in charge of lactate or pyruvate flux, and PPARα, a TF closely linked to lipid metabolism. The interaction with these metabolic effectors favors not only glycolysis, but also improves mitochondrial oxidative phosphorylation and the β-oxidation of fatty acids, allowing urothelial tumor cells to use alternative energy sources, which translates into metabolic plasticity marked by a hybrid glycolysis/OXPHOS phenotype known to facilitate BC invasiveness.

## 4. Non-Invasive Bladder Cancer Biomarkers

A biomarker could be defined as a characteristic which is objectively measured and capable of indicating the state of a biological process, be it normal or pathological [219]. In cancer, and specifically BC, diagnostic, prognostic, and monitoring biomarkers have been identified in biological samples, e.g., blood and urine, by high-throughput techniques [220]. In this section, we review the most important metabolomic and epigenomic studies carried out in the context of BC.

### 4.1. Metabolomic Studies in BC

Metabolomics identifies and quantifies endogenous and exogenous low-molecular-weight organic molecules (<1 kDa), i.e., metabolites, that are present in a biological sample [221]. Analytical platforms such as gas chromatography (GC) and liquid chromatography (LC) coupled to mass spectrometry (MS) and nuclear magnetic resonance (1H NMR) spectroscopy have been widely used in metabolomic analyses to identify potential oncological biomarkers [220,221]. Each of these techniques has advantages and limitations related to sample preparation, detection range, analysis speed, thresholds of sensitivity and specificity, so the choice of either one of these approaches depends on the aims and the requirements of the study.

In the last decade, metabolomic analyses have provided metabolomic profiles intended to be used in BC surveillance, or capable of distinguishing: (i) BC from control samples; (ii) NMIBC from MIBC samples; or (iii) low-grade from high-grade BC, with sensitivities, specificities, positive and negative predictive values (PPV, NPV) over 75 to 80%, as well as an elevated area under the curve (AUC). Although some of these studies have only provided holistic profiles [25], others have identified potential metabolites which may be used as non-invasive biomarkers. Table 1 and Table 2 summarize the main metabolites identified through different metabolomic platforms in urine and serum samples, as well as the biochemical pathways in which they are involved. 

Figure 3 shows the results obtained after performing an analysis of altered metabolic pathways in BC using the MetaboAnalyst 3.0 tool, and considering all the found discriminant metabolites both in urine as in serum samples. Among the set of identified metabolites in BC, a large majority are linked to pathways related to amino acid metabolism, the TCA cycle, or pyruvate metabolism. These data derived from studies performed in BC samples [14,225] share altered metabolic pathways with other types of tumors [17], highlighting that metabolic reprogramming is a common hallmark of tumors.

It is important to note that all these discriminant metabolites were identified in studies where tumor and non-tumor samples were compared, so clinically they could be applied as diagnostic biomarkers. Conversely, other studies identified metabolites related to BC aggressiveness using samples from LG or HG tumors [44,234,235,236]. For instance, Bensal et al. identified a serum metabolic profile composed of six metabolites (dimethyl amine (DMA), malonate, lactate, glutamine, histidine, and valine) able to distinguish LG and HG BC samples from control samples. All except malonate were identified as crucial to segregate LG tumors from controls; DMA, malonate, lactate, and histidine served as differentiating biomarkers of HG from controls and the combination of DMA, glutamine, and malonate was sufficient to accurately segregate LG from HG [236]. Tan et al. also performed a metabolomic study using serum samples of patients with LG and HG BC. In this case, they found a panel of serum metabolites formed by the combination of inosine, N-Acetyl-N-formyl-5-methoxykynurenamine (AFMK) and PS(O-18:0/0:0) which sufficiently discriminated not only HG BC and LG BC (AUC > 0.95), but also LG BC and healthy controls (AUC ≈ 0.99) [235]. The last notable study was carried out by Zhou et al. They observed that both HG and LG BC had distinct metabolic profiles when compared to control samples (e.g., elevated concentrations of TCA cycle metabolites or fatty acid biosynthesis metabolites). Additionally, HG tumors had higher levels of PPP intermediates, nucleotide metabolites, and amino acids than the control group. Minor differences were detected in high- and low-grade tumors, with oleic acid and serine being identified as discriminant metabolites between both groups [44]. Overall, these data suggest that differential metabolic alterations are linked to tumor aggressiveness, which allows the modulation of the processes involved in the development and progression of BC (proliferation, immune escape, differentiation, apoptosis, and invasion). Finally, other studies have identified urinary metabolic profiles as biomarkers for BC monitoring [14,228]. In these studies, samples were sequentially collected from NMIBC patients undergoing long-term disease surveillance. Metabolic profiles were able to detect recurrences in this cohort of patients by being able to observe that, after tumor removal, the metabolic profile trajectory changed towards a non-tumor phenotype in concordance with negative cystoscopy results. Metabolites linked with tryptophan, phenylalanine, arginine, proline, taurine, and hypotaurine metabolic pathways were identified as discriminant and were postulated as potential biomarkers for BC monitoring.

In conclusion, although some metabolomic studies have identified potential metabolic profiles associated with BC, their transfer to the clinic remains challenging. With further research and validation, metabolomic profiles may acquire FDA or European Medicines Agency approval as diagnostic, prognostic or monitoring biomarkers. To achieve this, the following are required: (i) the establishment of standard protocols that guarantee sample quality during collection and processing; (ii) control of the analysis quality in order to reduce preanalytical variation and batch-to-batch variability of data; (iii) a greater reproducibility among metabolites found in similar studies; and (iv) the validation of metabolic profiles in large patient populations.

### 4.2. Epigenetic Biomarkers in BC

The great advantage offered by epigenetic biomarkers is that they are highly dynamic, showing a reversible and measurable expression in the different stages and grades of tumors. Liquid biopsies more vastly used in the detection and surveillance of BC have used urine and serum/plasma as biofluids. These liquid biopsies present several epigenetic biomarkers, such as ctDNA for DNA alterations (mutations, CNV, methylation) or miRNAs and lncRNAs, and the detection of exosomes (microvesicles that protect small RNAs that can be found in the urine of patients).

Some epigenetic enzymes participate in metabolic reprogramming and may be considered good therapeutic targets, but they are not good biomarkers in fluids because: (i) there are other ways to measure them; (ii) there are not enough validation studies to support them as biomarkers, either for diagnosis, prognosis, recurrence or follow-up. Some of these, such as EZH2 or G9A, are good molecular markers in bladder tumor tissue, which can help to discern the nature of a tumor subtype and the prognosis of patients [116] and can also cooperate or regulate other epigenetic elements which are biomarkers in the clinic. The same situation occurs with some miRNAs and LncRNAs such as miR-143, miR-145, miR-200, miR-34a, UCA-1 or HOTAIR. They are good biomarkers in tissue expression, but they are not validated in fluids. Thanks to massive sequencing techniques, molecular panels have been achieved that mark patterns in bladder tumor subtypes [63], a fundamental tool for the classification of patients and their therapeutic opportunities [5].


**DNA Methylation**


Pyrosequencing techniques and comparative analysis of large databases allow us to discover DNA methylation events, also important in tumor metabolism. The hypermethylation of gene promoters occurs in 50–90% of BCs and includes a series of genes that are considered biomarkers, either in urine or serum. Some of them are *APC, ARF, BAX, BCL2, CDH1, CDKN2A, DAPK, EDNRB, EOMES, FADD, GDF15, GSTP1, LITAF, MGMT, NID2, PCDH17, POU4F2, RARβ2, RASSF1A, TCF21, TERT, TIMP3, TMS-1, TNFRSF21, TNFRSF25*, and *ZNF154* [228,237]. Many of them have been validated in large cohorts of patients and have reliable values for sensitivity, specificity and ROC curves.

Among all the alterations in the epigenetic machinery discussed previously, we must distinguish that they exist as validated biomarkers for BC, but whose implication in metabolic reprogramming has not been studied or demonstrated in BC yet. We highlight studies such as Hauser et al., who reported *TIMP3, APC, RAR-β2, TIG1, p16INK4a, PTGS2, p14ARF, RASSF1A* and *GSTP1* methylation promoters on the cell-free serum of BC patients for diagnosis [99,237]. The combination of methylated *CDKN2A, GSTP1* and *MGMT*, enzymes related to DNA repair events and drug metabolism [233,238], achieves 70% sensitivity and 100% specificity in BC detection [239]. *GSTP1* and *MGMT* in combination with *CDKN2A* and *ARF* have diagnostic power in the urine of patients with BC [239], and *GSTP1* together with *TIMP3* promoter methylation allowed the discrimination of invasive tumors [13,239]. Therefore, the identification of these genes within a population may help to better identify genetic vulnerabilities and pharmacogenetic studies and also monitoring of patients during chemotherapy. However, the cohorts of patients are not enough in some of these studies. In addition, DNA methylation patterns represent marks that can be detected throughout the development of the tumor and its progression. As the bladder tumor becomes malignant and invasive (MIBC), there are more alterations in DNA methylation, which can be used as a clinical prognostic tool. Regarding its connection with metabolism, on many occasions, we find genes associated with xenobiotic metabolism, which could give us information on resistance to previous therapies. Findings on methylation biomarkers in BC patients together with these are summarized in Table 3.


**ncRNAs**


The main epigenetic markers that can be found in liquid biopsy are miRNAs and lncRNAs. They are stable molecules that, due to their expression levels, can be related to grade, state and other characteristics of the tumor. Thus, some of them are already diagnostic (tumor initiation), prognosis (tumor in development or progression), or follow-up markers (recurrent tumor/progression/dissemination).

Most of the biomarkers described and validated for diagnosis or prognosis in patients with BC are ncRNAs, and some of them are related to the regulation of factors involved in tumor metabolism. The combination of lncRNAs offers more precise results. We underline the possible connections between epigenetics and metabolism (Table 3, Table 4, Table 5 and Table 6) from BC biomarkers summarized by Lodewijk et al. [13]. Among detectable miRNAs and lncRNAs in liquid biopsy, those highlighted are or could be involved in the regulation of metabolic reprogramming in BC. We emphasize the expression of the miR-200 family, miR-21, miR-34a, miR-143 or miR-93. We can highlight that the cluster of miRNAs detected in urine for recurrence and surveillance reflects a greater number of alterations related to enzymes and metabolic pathways than the sets of miRNAs used in diagnosis (Table 4). Thus, the metabolism could be more altered as the disease progresses, and recurrence processes would represent a period of great metabolic changes for the cell. 

On the other hand, the downregulation of miRNA expression could be a characteristic feature of invasive tumors. Some examples are miR-200, miR-1, miR-143, miR-145, miR-133a, miR-133b, and miR-125b [13], which have a tumor suppressor role and their silencing promotes reprogramming and EMT processes, and also most of them are associated with metabolic reprogramming in recurrence and xenobiotic metabolism, suggesting that their regulation program is active in more advanced stages of the disease. It is important to note that the particular role in metabolic reprogramming of ncRNAs detected in urine and in the validation sets from MIBC serum sample assays (Table 4 and Table 5) has not been discovered yet.

Regarding lncRNAs, there are very few validated ones, but we can highlight a validated panel from urine samples which combines the expression of hyaluronoglucosaminidase 1 (HYAL1), miR-210, miR-96 and lncRNA UCA-1 (Table 4), thereby achieving a sensitivity of 100% and a specificity of 89.5% as a diagnosis biomarker [240]. LncRNAs such as UCA-1, MALAT1 or HOTAIR would be of great interest to carry out studies in combination of some of them with SIRTs, since there is evidence that they work in coordination in metabolic rewiring on some occasions, as we have commented previously.

In addition, miRNAs and lncRNAs are generally included in exosomes, which are crucial for the communication of cancer and stromal cells. Exosomes preserve their integrity and are very stable in liquid biopsy samples, such as serum, plasma, and urine, making them potential diagnostic and prognostic biomarkers with non-invasive methods [13]. We report a study using exosomes from the urine of patients with MIBC that carry lncRNAs (HOTAIR, HOX-AS-2, MALAT1, HYMAI, LINC00477, LOC100506688 and OTX2-AS1) as diagnostic biomarkers [241] (Table 6). The identification of exo-miRNAs associated with metastases could provide an additional tool to evaluate the follow-up of progression disease [242].

Finally, it should be mentioned that both metabolites as ncRNAs or methylation patterns in DNA are potential biomarkers that can be translated into the clinic to improve the diagnosis, prognosis and monitoring of BC. All these biomolecules provide dynamic information about tumor biology and evolution from a non-invasive and cost-effective approach.

## 5. Therapeutic Opportunities and Future Perspectives

Massive sequencing technologies have allowed us to advance in the management of BC, being a fundamental tool for the stratification of patients by increasingly well-known molecular subtypes. Besides, the study of omics has given a greater vision of the behavior of a tumor and its dynamics as stages progress. Thanks to studies in the field, new therapeutic options are opened. This information, together with that provided by biomarkers, makes it possible for us to talk more about precision medicine applied to patient management and clinical decision-making. Nevertheless, it is difficult to ensure that emerging therapies are implemented early in all patients, although some of them show promising results. 

Clearly, a key metabolic target point is the glycolytic pathway and TCA cycle. There are potent blockers of several metabolic pathways such as GLS inhibitors, competitive G6P analogs that decrease acetyl-CoA levels [21,27], glucose transporter inhibitors [74] and even hexosamine biosynthesis pathway inhibitors that can decrease protein O-GlcNAcylation levels and reverse glucose-mediated metabolism [21]. Recent studies also suggest SAM and SAH inhibitors as potential antitumor candidates and even methionine-restricted diets [21,27,118]. SAH not only regulates intracellular levels of SAM, but the availability of SAM is critical for DNMT and HMT activity [21,27,86]. A well-known example is DzNep, an SAH inhibitor used in BC that blocks EZH2 and is closely related to H3K27me3 levels [199]. The levels of α-KG also affect the methylation status, as it is an essential cofactor for DNA demethylase (TET) and histone demethylase (JMJD) and could be affecting the levels of H3K27me3 [10]. In BC, there seems to be a positive feedback between the levels of SAM, α-KG and HMTs, so the use of inhibitors of these metabolites could lead to a promising novel strategy in this disease.

The main problem in the use of metabolic inhibitors is the heterogeneity in the metabolic profiles of the tumor cells, allowing them to escape the pharmacological effect [73]. Therefore, it is important to have information on the metabolic data from patients when selecting these inhibitors.

Regarding epigenetic therapy, two therapeutic strategies are currently used: small molecules that inhibit epigenetic enzymes and the manipulation of the expression and activity of miRNAs [243]. Epigenetic inhibitors currently approved for the treatment of cancer primarily target DNA methylation (anti-DNMTs) and histone modifications (anti-HDACs) [244].

The latest studies have focused on determining the expression pattern of HDACs in different cell lines and bladder tumors to explore their role in the development of cancer and to be able to predict the efficacy of drugs [245]. Although HDAC inhibitors are shown to be successful clinically, none of them targets class III HDACs. 

Regarding SIRTs, recent studies have tried to identify SIRT6 inhibitors to understand its mechanism of action [21,22]. Other reports have shown that the inhibition of SIRT7 reduces tumor growth, and the deacetylation of its mark promotes the inhibition of miR-34a expression [150], implicated in BC. However, although there are many known compounds that regulate the activity of SIRTs, their use as therapeutic drugs is uncertain. SIRTs regulate so many cellular processes, so a change in their activity could positively affect one face of the disease but have a negative impact on the other. However, there is intense and ongoing research on its therapeutic use.

On the other hand, new epigenetic inhibitors targeting other histone remodeling enzymes (HDMTs, HMT, etc.) are being developed. These include improved DZNep analog blockers, specifically targeting EZH2 [246], G9A inhibitors [247] or agents against other HMTs [248]. Dual inhibitors are more effective, such as EZH2/EZH1 (UNC1999) [249], or against G9a/DNMT1 (CM-272) [116,250], which could represent an improved approach in cancer therapeutics, especially in BC. One of the great problems of EZH2 as a target is its relationship with acquired drug resistance. Therefore, it would be interesting to study the possible relationship between EZH2 and xenobiotic metabolism genes. EZH2 is a powerful regulator of gene expression, in addition to having other non-canonical roles, leading to the activation of different signaling pathways to maintain tumor cell viability [128]. Furthermore, EZH2 can interact with other non-histone proteins that regulate its activity by phosphorylation, such as AMPK or AKT (149, main activators of glycolytic metabolism). Therefore, it is important to develop therapeutic strategies based on the potential value of the combined intervention of EZH2 and tumor metabolic activities.

Synthetic lethality arises as a very attractive approach, based on the loss of expression of two antagonistic genes, as occurs with KDM6A (loss of expression mutation) and EZH2 (sensitization to inhibitors) in BC [148]. This strategy improves the knowledge of signaling pathways both to define with greater precision the different subtypes of tumors and to understand the drug-resistance mechanisms.

As mentioned at the beginning, there is another focus of epigenetic therapy aimed at miRNAs. Antisense oligonucleotides have been developed for silencing and synthetic miRNAs or lentiviral constructs are used to restore expression [251,252]. There are basic studies where the use of lentiviruses and antisense oligonucleotides, in combination with cisplatin or other chemotherapeutic agents, increases the sensitization of BC cells [253,254]. The use of synthetic RNAs to increase/decrease expression is also increasing. For example, the downregulation of miR-34a causes a clear inhibition of the clonogenic potential [255], restoring miR-143, and it can target HK2 [21], or miR-101 overexpression, which enhances the sensitivity to treatment [164]. On the other hand, oncogenic miRNAs that control tumor suppressive pathways (LKB1, AMPK, PTEN…), such as miR-21, can be inhibited and allow TSG re-expression [21]. Recently, the CRISPR/Cas9 genome editing technique had been employed to knockout lncRNA UCA-1 (147). Nevertheless, the development of this type of therapy raises several challenges such as reliable administration methods, the determination of appropriate dosages, as well as the possible pleiotropic effects or resistance to therapy [165,192,194]. 

The ncRNAs exert an extensive and complex influence on the metabolic networks that characterize reprogramming in tumors, but currently, they mostly remain uncharacterized. Therefore, deepening the investigations of their functions and mechanisms in the regulation of metabolism is essential for the development of clinical therapies focused on patients with tumors with altered metabolism and the identification of new future biomarkers [193,194].

Apart from their possible therapeutic use in regulating the energy of cancer metabolism [165,192], the true potential application of ncRNAs lies in the growing interest in their biomarker character. Expression signatures of these ncRNAs in tumors hold great promise for the development of non-invasive diagnostic and prognostic biomarkers, which together with metabolome studies translates into a more powerful and sophisticated strategy for cancer diagnosis and treatment [192]. Additionally, exosomal-ncRNAs have several advantages over the existing approaches, such as low toxicity and target specificity [242].

The key questions to be solved in the near future are, on the one hand, that a stratification of patients must be carried out to guarantee the safety of the drugs. The metabolic and epigenetic landscape changes at various stages of tumor growth, making it difficult to design drugs that are effective [90]. Simultaneously targeting the epigenetic and metabolic pathways through combined therapies can inhibit the dynamic adaptive mechanisms of tumor cell reprogramming and obtain synergistic effects that can be evaluated in clinical studies [20].

On the other hand, resistance to therapy or a lack of pharmacological efficacy leads us to a second limitation, intertumoral and intratumoral heterogeneity—intertumoral because the pharmacodynamics of each patient tumor is different, intratumoral because the subpopulations that can be found within the tumor present different epigenetic and metabolic profiles that are regulated differently and complicate therapeutic interventions [21,73,90]. Against this, emerging and rapidly advancing tools and platforms to study epigenomic and metabolomic single cell heterogeneity are allowing the isolation and distinction of heterogeneous subpopulations of cancer cells that help us to understand the different profiles that we can find. Liquid biopsies have also been proposed as an efficient and accessible way to decipher the intratumor heterogeneity of the patient, by analyzing circulating tumor cells (CTC) and extracting circulating tumor DNA (cDNA) [21,90]. Therefore, future clinical trials should incorporate the analysis of epigenomic and metabolomic biomarkers that allow the selection of subsets of patients who may benefit from available treatments [21]. In the case of BC, although multiple genomic analyses have been carried out to study alterations in metabolic enzymes, there are no reports that integrate transcriptomic and metabolomic analyses from any type of biopsy until now [14]. 

Metabolites and epigenetic molecules as biomarkers are able to record and monitor the stage of the disease in real time and predict prognosis, recurrence or progression, as well as the evaluation of response to treatment and resistance. However, to demonstrate the reliability of a biomarker and the positive impact, a reliable prospective validation is unavoidably required, in addition to the need for large studies with long follow-up periods and large cohorts of patients [13,256,257]. Thus, in BC, these monitoring systems could be used as a new approach to achieve unequivocal diagnoses and minimize the invasiveness of the tests within its handling and management framework.

## 6. Conclusions

The metabolome and epigenome are closely intertwined in BC. EZH2 interacts with chromatin remodelers such as G9a and DNMTs, but also with HIF, MYC, and oncogenic signaling cascades, such as PI3K/Akt, to regulate metabolic reprogramming. Specifically, it appears that this regulation enhances the synthesis of α-KG from glucose and glutamine catabolism and promotes glucose flux towards glycolysis, PPP, and lipogenesis processes. Additionally, data suggest an important role of amino acids, or their metabolite derivatives, in BC, such as serine, methionine, SAM, and SAH, which are involved in one-carbon metabolism, as well as oncometabolites such as succinate, fumarate, and α-KG involved in the TCA cycle. All these metabolites have a crucial role in acting as epigenetic cofactors or substrates, but also in inhibiting or activating epigenetic enzymes that control the chromatin state and therefore gene expression. In summary, these data show the need for further research in this promising field to offer patients: (i) personalized treatments that increase their life expectancy; (ii) and non-invasive bladder cancer diagnosis and monitoring techniques that improve their quality of life, being cost-effective for health systems.

## Figures and Tables

**Figure 1 cancers-13-02719-f001:**
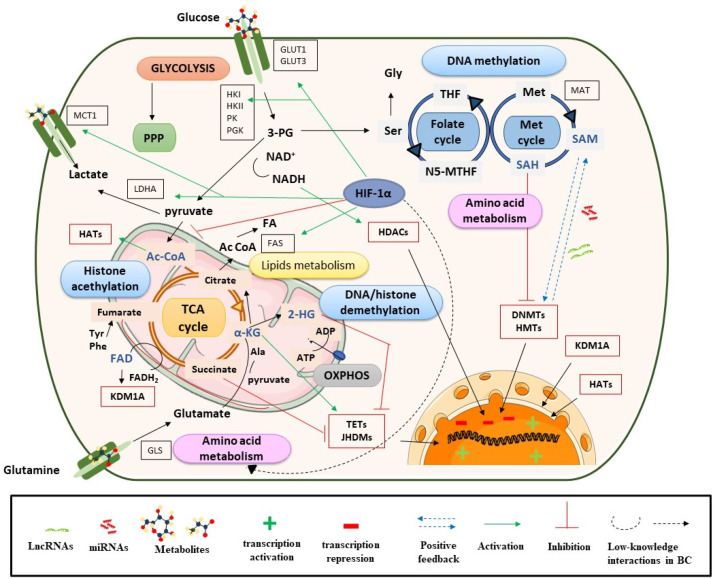
Metabolism controls epigenetic enzymes in BC. Ac CoA: acetyl coenzyme A; ADP: adenosine diphosphate; α-KG: alpha-ketoglutarate; ATP: adenosine triphosphate; DNMTs: DNA methyltransferases; FADH2: flavin adenine dinucleotide; GLS: glutaminase; GLUT: glucose transporter; HADAC: histone deacetylase; HATs: histone acetyltransferases; 2-HG: 2-hydroxyglutarate; HIF-1α: hypoxia-inducible factor subunit alpha; HMTs: histone methyltransferases; HK: hexokinase; JHDMs: JmjC-domain-containing histone demethylases; KDMs: lysine demethylases; LDHA: lactate dehydrogenase A; Met: methionine; NADH: nicotinamide adenine dinucleotide; N5-MTHF: 5-methyltetrahydrofolate; OXPHOX: oxidative phosphorylation; 3-PG: 3-phosphoglycerate; PGK: phosphoglycerate kinase; PI3K: phosphoinositide 3-kinases; PK: pyruvate kinase; PPP: pentose phosphate pathway; SAM: S-adenosyl methionine; Ser: serine; TCA cycle: tricarboxylic acid cycle; TETs: ten eleven translocation enzymes; THF: tetrahydrofolate.

**Figure 2 cancers-13-02719-f002:**
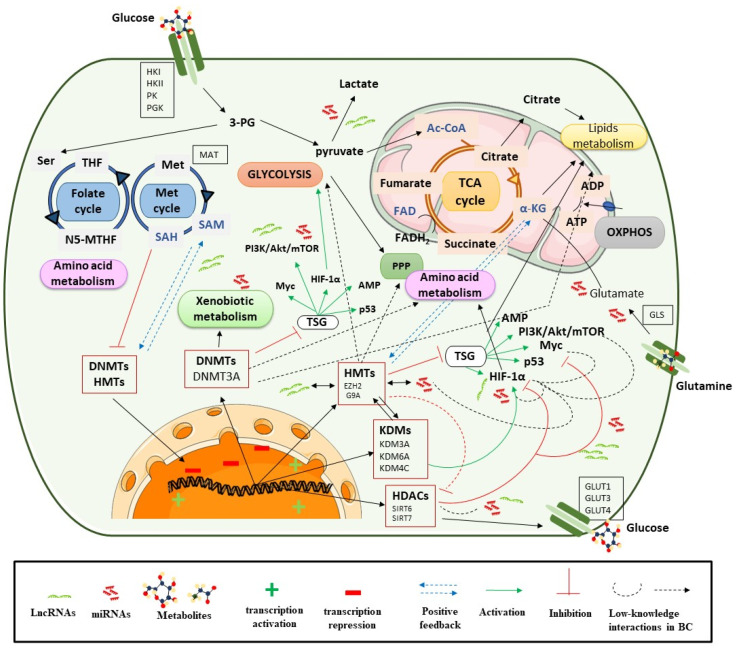
Epigenetic processes control metabolic pathways. Ac CoA: acetyl coenzyme A; ADP: adenosine diphosphate; Akt: alpha serine/threonine-protein kinase; α-KG: alpha-ketoglutarate; ATP: adenosine triphosphate; DNMTs: DNA methyltransferases; EZH2: enhancer of zeste homolog 2; FADH2: flavin adenine dinucleotide; G9A: euchromatic histone-lysine N-methyltransferase 2; GLS: glutaminase; GLUT: glucose transporter; HDAC: histone deacetylase; HAT: histone acetyltransferase; 2-HG: 2-hydroxyglutarate; HIF-1α: hypoxia-inducible factor subunit alpha; HMT: histone methyltransferase; HK: hexokinase; JHDMs: JmjC-domain-containing histone demethylases; KDM: lysine demethylase; LDHA: lactate dehydrogenase A; LncRNA: long non coding RNA; Met: Methionine; Myc: avian myelocytomatosis viral oncogene homolog; miRNA: microRNA; mTOR: mammalian target of rapamycin; NADH: nicotinamide adenine dinucleotide; N5-MTHF: 5-Methyltetrahydrofolate; OXPHOX: oxidative phosphorylation; 3-PG: 3-phosphoglycerate; PGK: phosphoglycerate kinase; PI3K: phosphoinositide 3-kinases; PK: pyruvate kinase; PPP: pentose phosphate pathway; SAM: S-adenosylmethionine; Ser: serine; SIRT: sirtuin; TCA cycle: tricarboxylic acid cycle; TETs: ten eleven translocation enzymes; THF: tetrahydrofolate; TSG: tumor suppressor gene.

**Figure 3 cancers-13-02719-f003:**
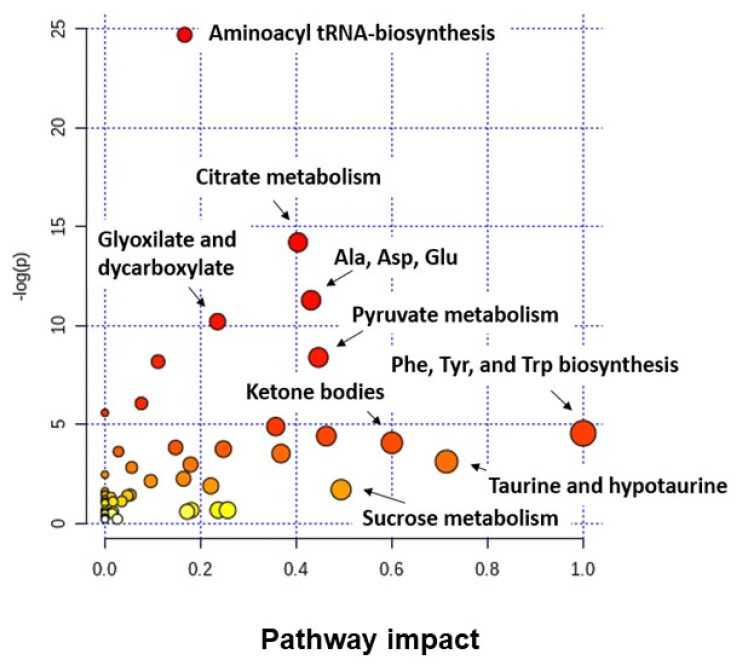
Analysis of altered metabolic pathways in BC when tumor and non-tumor samples (urine and serum) are compared. Note: the color and the size of each circle indicate its *p*-value and pathway impact value, respectively. Ala: alanine; Asp: aspartate; Glu: glutamine; Phe: phenylalanine; Tyr: tyrosine; Trp: tryptophan.

**Table 1 cancers-13-02719-t001:** Putative identified metabolites and associated pathways in BC urine samples.

Metabolic Biomarkers in BC			
Urine			
Perturbed Biochemical Pathway	Levels (BC/Control)	Metabolites	Clinical Application
Glycolysis	High	Fructose [222], lactic acid [26,223]	Diagnosis
	Low	Fructose [26]	Diagnosis
TCA cycle	High	--	--
	Low	Citric acid [222,223,224], succinate [26]	Diagnosis
Amino acid metabolism	High	Val [20,26,222], Phe, Met, S−Adenosylmethionine, Lys [225], Leu [26,225,226], Ile, His, Ser [26], Tyr, Trp, hydroxyphenylalanine, phenylacetilglutamine, homophenylalanine, phenylglycoxylyc acid, kynurenine, hydroxyhippuric acid [221,227], hydroxytryptophan, indolacetic acid, minohippuric acid [227]	Diagnosis
	Low	Ala, PAGN, Pro, Arg [226], Asp [225,226], hippuric acid [224,227,228], creatine [26,229,230]	Diagnosis
GSH metabolism	High	--	--
	Low	Pyroglutamic acid [231]	Diagnosis
Taurine and hypotaurine	High	Taurine [224,230,232]	Diagnosis
	Low	--	--
Lipid metabolism	High	Carnitine [225], acetylcarnitine [26,227,230]	Diagnosis
	Low	Glycerol [222], palmitic acid [225]	Diagnosis
Nucleotide/nucleoside metabolism	High	Thymine [227], hypoxanthine, uridine [222]	Diagnosis
	Low	Adenosine [26]	Diagnosis
NAD cycle	High	--	
	Low	Trigonelline [229]	Diagnosis

Note: This table only includes metabolites that were quantified individually in urine samples and obtained significant *p*_value (*p*_value < 0.05), or metabolites that were selected as discriminants in metabolomic studies in which a validation set was used and the model’s performance was good (i.e., sensitivity, specificity, PPV and NPV > 0.75). Ala: alanine; Asp: aspartate; Gly: glycine; GSH: glutathione reductase; His: histidine; Ile: isoleucine; Leu: leucine; Lys: lysine; Met: methionine; NAD: nicotinamide adenine dinucleotide; PAGN: phenylacetylglutamine; Phe: phenylalanine; Pro: proline; Ser: serine; TCA: tricarboxylic acid; Tyr: tyrosine; Val: valine; --: unknown.

**Table 2 cancers-13-02719-t002:** Putative identified metabolites and associated pathways in BC serum samples.

Metabolic Biomarkers in BC			
Serum			
Altered Biochemical Pathway	Levels (BC/Control)	Metabolites	Clinical Application
Glycolysis	High	Glucose [232], erythritol, D-lyxosylamine, ribonic acid [44]	Diagnosis
	Low	Lactate [232]	--
PPP	High	Ribose, gluconic acid, 2-keto-gluconic acid, xylitol, arabitol [44]	Diagnosis
	Low	--	--
Sucrose metabolism	High	Galacturonic acid, D-cellobiose, maltose [44]	Diagnosis
	Low	--	--
TCA cycle	High	Succinate, pyruvate, oxalacetate, phosphoenolpyruvate, acetyl-CoA [28], Cis-aconitic acid, fumaric acid, malic acid [44]	Diagnosis
	Low	Citrate [232]	Diagnosis
Amino acid metabolism	High	Gln, His, Malonate, Val [233], creatinine, kynurenine, norleucine [44]	Diagnosis
	Low	Tyr, Ile, Phe, Leu, Gly [232]	Diagnosis
Taurine and hypotaurine	High	Hypotaurine [44]	Diagnosis
	Low	--	--
Lipid metabolism	High	Carnitine [28]	Diagnosis
	Low	--	--
Nucleotide/nucleoside metabolism	High	Uridine, hypoxanthine [44]	Diagnosis
	Low	--	--
Organic acid	High	2-hydroxyglutaric acid, (R,R)-2,3-dihydroxybutanoic acid, 2,3,4-trihydroxybutyric acid, 2,4-dihydroxybutanoic acid, 3,4,5-trihydroxypentanoic acid, 3,4-Dihydroxybutanoic acid [44]	Diagnosis
	Low	--	--
Choline	High	Choline [232]	Diagnosis
	Low	--	--
Ketone metabolism	High	Acetoacetate [232]	Diagnosis
	Low	--	--

Note: This table only includes metabolites that were quantified individually in serum samples and obtained significant *p*_value (*p*_value < 0.05), or metabolites that were selected as discriminants in metabolomic studies in which a validation set was used and the model’s performance was good (i.e., sensitivity, specificity, PPV and NPV > 0.75). Gly: glycine; Gln: glutamine; His: histidine; Ile: isoleucine; Leu: leucine; Phe: phenylalanine; PPP: pentose phosphate pathway; TCA: tricarboxylic acid; Tyr: tyrosine; Val: valine; --: unknown.

**Table 3 cancers-13-02719-t003:** Epigenetic marker compilation related to gene methylation status of ctDNA found in urine and serum from patients.

ctDNA (Urine and Serum)				
Metabolic Gene/Pathway Related	Gene Status	Urine	Serum	Clinical Application
Xenobiotic metabolism	Hipermet.	*CDKN2A*, *MGMT ARF, GSPT1*	--	diagnosis
Xenobiotic metabolism, lipid metabolism and β-oxidation fatty acids	Hipermet.	*--*	*TIMP3, APC, RAR-β2, TIG1, p16INK4a, PTGS2, p14ARF, RASSF1A, GSTP1*	diagnosis
Unknown	Hipermet.	*GDF15, TMEFF2, VIM*	--	diagnosis
Unknown	Hipermet.	*PCDH10, PCDH17, APC*	--	prognosis
Unknown	Hipermet.	*TWIST, NID2*	--	diagnosis
Unknown	Hipermet.	*--*	*p16INK4a, p14ARF, CDH1, PCDH10, PCDH17*	diagnosis

Note: Those biomarkers that have been related to tumor metabolism in BC are highlighted in red.

**Table 4 cancers-13-02719-t004:** ncRNA markers found in urine from patient cohorts with BC.

Urine			
Metabolic Gene/Pathway Related	Levels	Biomarker Panels	Clinical Application
GLUT1, GLUT4LDHA, HIF1	High	miR-652, miR-199a-3p, miR-140-5p, miR-93, miR-142-5p, miR-1305, miR-30a, miR-224, miR-96, miR-766, miR-223, miR-99b, miR-140-3p, let-7b, miR-141, miR-191, miR-146b-5p, miR-491-5p, miR-339-3p, miR-200c, miR-106b, miR-143, miR-429, miR-222 and miR-200a	Diagnosis
Unknown	High	miR-7-5p, miR-22-3p, miR-29a-3p, miR-126-5p, miR-200a-3p, miR-375 and miR-423-5p	Diagnosis
GLS, LHDA, HIF1,GLUT1, GLUT3, LHD, PKM2, HK2LDHA, HK1 (p53)	High	UCA1-miR-16, miR-200c, miR-205, miR-21, miR-221 and miR-34a	Recurrence and surveillance
Unknown	High/low	**NMIBC**: miR-30a-5p, let-7c-5p, miR-486-5p, miR-205-5p and let-7i-5p **NMIBC** (high grade): miR-30a-5p, let-7c-5p, miR-486-5p, miR-21-5p, miR-106b-3p, miR-151a-3p, miR-200c-3p, miR-183-5p, miR-185-5p, miR-224-5p, miR-30c-2-5p and miR-10b-5p **MIBC**: miR-30a-5p, let-7c-5p, miR-486-5p, miR-205-5p, miR-451a, miR-25-3p, miR-30a-5p and miR-7-1-5	Diagnosis/prognosis
p53, HIF1	Low	miR-125b, miR-204, miR-99a, miR-30b, and miR-532-3p.	Diagnosis
Glutamine metabolism, xenobiotic metabolism, mitochondrial activity, HIF1	High	hyal, lncRNA UCA1, microRNA-210, microRNA-96	Diagnosis (MIBC)

NOTE: In red are indicated biomarkers related to metabolism. High expression of biomarkers is highlighted in green and low expression in blue.

**Table 5 cancers-13-02719-t005:** miRNA markers found in serum patient cohorts with BC.

Serum			
Metabolic Gene/Pathway Related	Level	Biomarker Panels	Clinical Application
Unknown	High	miR-422a-3p, miR-486-3p, miR-103a-3p and miR-27a-3p	Prognosis (MIBC)
Unknown	High	miR-152, miR-148b-3p, miR-3187-3p, miR-15b-5p, miR-27a-3p and miR-30a-5p	Prognosis
Unknown	High	miR-422a-3p, miR-486-3p, miR-103a-3p and miR-27a-3p	Prognosis (MIBC)
Unknown	High	miR-541, miR-200b, miR-566, miR-487 and miR-148b	Diagnosis
Unknown	Low	miR-25, miR-92a, -92b, miR-302 and miR-33b	Diagnosis
Unknown	High	miR-152	Prognosis

**Table 6 cancers-13-02719-t006:** lncRNA markers found in BC-derived exosomes from urine of patient cohorts.

lncRNA-Derived Exosomes (Urine)			
Metabolic Gene/Pathway Related	Levels	Biomarker Panels	Clinical Application
GLUT1	High	*HOTAIR, HOX-AS-2, MALAT1, HYMAI, LINC00477*	Diagnosis (MIBC)

NOTE: biomarkers related to metabolism are in red.

## Data Availability

No new data were created or analyzed in this study. Data sharing is not applicable to this article.

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
