# Peer review of "Epigenomic and Metabolomic Integration Reveals Dynamic Metabolic Regulation in Bladder Cancer"

_cancers, 2021, doi:10.3390/cancers13112719_

Round 1

Reviewer 1 Report

The review presented here is new information about disease diagnostics and tumor features in relations to cancer epigenomic, metabolism, and cell signaling pathways in the context of BC. It provides an overview of this enormous and complex interplay between these biochemical and molecular processes and highlights the metabolites, metabolic enzymes, miRNAs, and lncRNAs postulated as emerging clinical biomarkers and therapeutic targets in the context of BC. The information presented in this paper is very useful for diagnosis, prognosis, monitoring, or treatment in BC.

Comments: 

  1. Figure 1 looks very busy. It should break down into 2 figures. One illustrates metabolism controls epigenome. The other one illustrates epigenetic controls metabolism.
  2. The authors presented the new findings and studies about metabolism controls epigenome and epigenetic controls metabolism. However, the authors only presented facts but lack their own discussions and interpretations in details. Please revise extensively in this part. 
  3. Under 2.2, what are the connections/biological consequences between alterations in metabolic processes and histone acetylation and disease formation and progression. Please address the clinical relevance in this section. 
  4.  Under 2.3, The clinical outcome such as drug resistance and metabolites in the nucleus need to be expanded more. Please provide authors'own discussions and interpretations. 
  5.  Under 3.1, The authors need to discuss the relationship between alteration of DNA methylation and BC growth and progression. 

Author Response

Dear reviewer, we attach the response document to the suggestions and comments made on the manuscript. We hope that it is of interest to you and all doubts have been resolved.

Reviewer 2 Report

In the present review manuscript entitled "Epigenomics and metabolomics integration reveals a dynamic metabolic regulation in bladder cancer", the authors have summurized the current knowledge on the dynamic metabolic regulation of bladder cancer from the epigenomics and metabolomics perspective.

This is a very well-written manuscript on a very interesting and modern topic that deserves the review and the interpretation of the novel research findings. In this regard, the authors have made an extensive search of the current literature and they have prepare an up-to-day review article on the discussed topic.

In general is really easy to read and gain the most significant knowledge on the topic. In this regard, I have only to propose minor comments

  1. I believe that Figure 1 is really informative, but at the same time very complex for the readers. Therefore, I recommend to split the figure.
  2. The authors have focused on non-invasive diagnostic biomarkers in ncRNAs. However, bladder cancer prognosis, especially in NMIBC, is of first clinical interest. In this regard, the authors have to include a small paragraph of miRNAs and lncRNAs clinical significance in bladder cancer prognosis, including circulating and tumor-related markers. In order not to significantly increase the text, the authors could focused on miRNAs/lncRNAs with already discussed diagnostic significance (e.g. miR-143/145, miR-200, miR-34a, UCA1, H19, HOTAIR, GAS5).
  3. The authors should complete the las page of the manuscript (authors contribution etc.)
  4. Authors should consider include affiliations in English.   

Author Response

(The authors gave the same response as above.)

Reviewer 3 Report

Research over the last decade has highlighted intense interactions between intermediary metabolism and epigenetic regulation which are highly relevant to cancer in general. Loras et al. aim at reviewing the current state of knowledge on these various interactions in bladder cancer. The result is unfortunately disappointing. Following an introduction to the topic, the review contains sections on influences of metabolism on the epigenome and on the reverse relationship, which are followed by sections on diagnostic applications and therapeutic possibilities. However, the structure is poorly adhered to (to give one example of many, the section on methyltransferases rapidly deviates to regulation of HIF1) and the promised analysis of dynamic interactions is nowhere to be found. Overall, the manuscript is confusing, provides few clear insights and little reliable information and makes very difficult reading not only for the above reason, but several others as well. In detail:

  1. Even though these topics per se have been reviewed previously, it would have been very helpful to provide first an overview of epigenomic (and epigenetic) alterations as well as of metabolic alterations in bladder cancer, before delving into the mutual relationships. In the present manuscript, this information is introduced in a piecemeal and often redundant fashion throughout the manuscript.
  2. The authors should focus on bladder cancer and minimize the treatment of other cancer types.
  3. This focus would help to make referencing more comprehensive on the actual topic. In a review article, referencing is essential; many references in the present manuscript are incomplete, incorrect and often not representative.
  4. Especially the section on diagnostics emphasizes miRNAs. However, miRNAs are not strictly epigenetic regulators (rather posttranscriptional regulators of gene expression). In addition, several previous reviews have addressed the expression, functions and diagnostic applications of miRNAs in bladder cancer. This topic is therefore of peripheral interest here. Certain lncRNAs are involved in epigenetic or metabolic regulation, but the section on these relationships is very superficial.
  5. The authors need to check the manuscript for accuracy and factual correctness throughout. There are many inaccurate or outright false statements. Even the tables contain many mistakes (glutamine is not a gene, to name a striking example). This problem is exacerbated by many mistakes in grammar, semantics and style. Thorough editing for contents as well as language correction, preferably by a native speaker, are urgently required.
  6. The abstract is not very informative and in particular, does not fit with the title.

Author Response

(The authors gave the same response as above.)

Round 2

Reviewer 1 Report

The authors have revised the manuscript by following the comments and suggestions from the Reviewers. 

I view that the revision is well done. I have no further comments.

Reviewer 3 Report

The revised version of the review by Loras et al. has been improved by language editing, but still suffers from the major problems that I identified in my previous report. In brief, the scope is too broad, especially by including long sections on diagnostic applications of DNA methylation changes and miRNAs and on potential therapies (which are largely speculative at this point, as far as BC is concerned). From the present version, it also becomes evident that the authors do not distinguish between epigenetics and (adaptive, short-term) gene regulation. Tellingly, the concept of epigenetics is only introduced on p. 8. There are still too many references to other tumor types, and many of the conclusions are based on inferences from these and speculations that interactions between metabolism and epigenetic regulation will be similar in bladder cancer. However, in many cases the data for this cancer type is (regrettably) simply not there. There are a number of pertinent studies cited and in my opinion the authors would have been well advised to review these in more detail. Likewise, the section on (urinary and serum) metabolites as diagnostic markers of BC is quite interesting, but it does not fit with the scope indicated by the title. Unfortunately, there are still too many statements that are vague, misleading, imprecise or false.* References are now cited according to standard, but still a number are wrong and in other places where references would be needed none are provided. Referencing for those issues with which I am more familiar is inconsistent and appears quite arbitrary. For instance, a considerable number of large-scale studies on DNA methylation have been performed over the last decade, but only candidate analyses are cited, of which several have not stood the test of time. EZH2 is treated in several places and HATs and KDMs are mentioned, but the fact that two HATs, KMT2C/D and KDM6A are mutated in (taken together) almost all BC cases does not appear at all, although this is a main reason why epigenetics is thought to be so important in BC. Finally, the authors have unfortunately not followed my advice to first give an overview of epigenetic and metabolic alterations in BC, before dealing with their mutual interactions. Reading the revised version, I still think that this would have helped to better organize the manuscript and to clarify many arguments.

*One exemplary (and representative) page with comments is attached.
